# No More, No Less: Least-Privilege Language Models

**Paulius Rauba** [1]  **Dominykas Seputis** [2 3]  **Patrikas Vanagas** [4]  **Mihaela van der Schaar** [1]

## Abstract

Least privilege is a core security principle: grant each request only the minimum access needed to achieve its goal. Deployed language models almost never follow it, instead being exposed through a single API endpoint that serves all users and requests. This gap exists not because least privilege would be unhelpful—deployments would benefit greatly from reducing unnecessary capability exposure. The real obstacle is definitional and mechanistic: what does "access" mean inside a language model, and how can we enforce it without retraining or deploying multiple models? We take inspiration from least privilege in computer systems and define a class of models called *least-privilege language models*, where privilege is *reachable internal computation* during the forward pass. In this view, lowering privilege literally shrinks the model's accessible function class (as opposed to denying access via learned policies). We formalize deployment-time control as a monitor–allocator–enforcer stack, separating (i) request-time signals, (ii) a decision rule that allocates privilege, and (iii) an inference-time mechanism that selects privilege. We then propose *Nested Least-Privilege Networks*, a shape-preserving, rank-indexed intervention that provides a smooth, reversible control knob. We show that this knob yields policy-usable privilege–utility frontiers and enables selective suppression of targeted capabilities with limited collateral degradation across various policies. Most importantly, we see this as a defense of a completely new deployment paradigm which challenges the premise that we can only have output-level control of language models.

[1]University of Cambridge [2]Vinted, Vilnius, Lithuania [3]University of Amsterdam, Amsterdam, Netherlands [4]Baltic Institute of Advanced Technology, Vilnius, Lithuania. Correspondence to: Paulius Rauba <pr501@cam.ac.uk>.

*Proceedings of the $43^{rd}$ International Conference on Machine Learning*, Seoul, South Korea. PMLR 306, 2026. Copyright 2026 by the author(s).

## 1. Introduction

> ▶ *"The LLM provided simple instructions for how to cultivate Yersinia pestis, the bacterium that causes plague."*
>
> ▶ *"The potential for an unknown, grave biological threat propelled or even generated by LLMs cannot be ruled out."*

The above are two citations, verbatim, from the results of a red-team study on the operational risks of AI in large-scale biological attacks (Mouton et al., 2024). The fact that language models can provide information that is dangerous is by now well understood. While we have made progress with safeguards (Wang et al., 2024), we currently must accept a daunting prospect: over a billion people use language models every day (Kemp, 2025), and these models possess, *in principle*, access to information that can pose harm to the overall population.

The reason this is dangerous is that, prior to the era of language models, access to such information was highly safeguarded. Developing dangerous biological weapons, for example, typically required an extremely high level of educational attainment, possibly a PhD in chemistry, which functioned as an implicit barrier to adoption. That barrier is now vanishing. The key risk of language models leaking information is not that this information does not exist elsewhere (it clearly does), but that it can be explained relatively easily to anyone, with step-by-step instructions that remove these educational barriers. As a result, we face an extremely uncomfortable situation: if language models provide dangerous information to a malicious actor, it can be used against the public in life-threatening ways. While the risk may be small in any single interaction, with over a billion people using language models, the cumulative risk compounds rapidly.

What can we do about this? Three approaches have dominated the community's efforts to address this issue. **Firstly**, numerous training- and post-training-time methods to improve alignment have been developed. These include removing harmful data from training sets (O'Brien et al., 2025; Li et al., 2025), using human feedback as a signal to make models helpful via reinforcement learning (Bai et al., 2022a; Kaufmann et al., 2024; Dai et al., 2023) or related techniques (Rafailov et al., 2023), and safety-specific fine-tuning (Choi

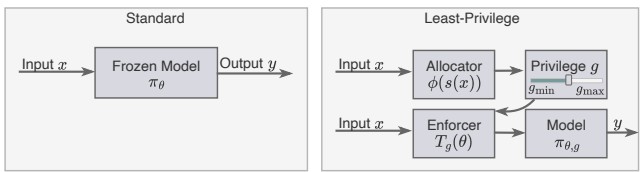

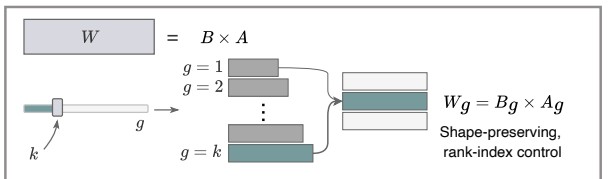

*(a)* **(A) Deployment-time control via least privilege.** Request-time signals $s(x)$ (risk, uncertainty, telemetry) are mapped by a privilege allocator $\phi$ to a control $g$. A capability enforcer applies $T_g$ *inside the forward pass*, inducing an effective deployed policy $\pi_{\theta,g}$ while preserving the external interface. The key shift is from output filtering to explicit control over which internal computations are reachable.

*(b)* **(B) Shape-preserving, rank-indexed enforcement.** Internal linear maps are re-parameterized with a nested factorization so that a deployer-set knob $g$ selects a prefix of the factors. This yields a monotone family $W(g)$ with $\text{rank}(W(g)) \leq g$ and fixed tensor shapes, recovering the baseline computation at $g_{\max}$. The result is a reversible, fine-grained inference-time control interface compatible with pretrained models.

*Figure 1.* **Least-privilege inference as an internal control problem. (A)** At deployment, the model parameters are fixed, but the amount of internal capability exercised need not be. Separating *signals*, *allocation*, and *enforcement* exposes a concrete action space for controllers: choose how much computation the model is allowed to use for each request. **(B)** We instantiate enforcement with rank-indexed, shape-preserving interventions that operate inside the forward pass. Together, these panels define a deployer-facing control surface on which allocation policies can be evaluated by utility, privilege cost, and runtime overhead.

et al., 2024), among others. **Secondly**, many output-level controls have been proposed. These include simple prompting strategies (Zheng et al., 2024), building "constitutions" (Bai et al., 2022b), or using rule-based methods to repair outputs deemed risky (Han et al., 2025), to name a few. **Thirdly**, there are activation-level approaches, such as activation steering (Turner et al., 2024; Stolfo et al., 2024), which enable control ("steering") of model behavior at deployment time.

However, such approaches are not fully satisfying. Superficially, we clearly observe that the risks still exist even in the presence of these approaches; that is, we have not yet solved the problem of language models revealing dangerous or unwanted information. Even more importantly, we are still left with the core problem.

> *Even after training-time alignment, output filtering, and activation-level steering, the base model still retains the underlying capability, because the relevant computations remain encoded in its weights and can still be activated during inference under some prompts or sampling.*

Can we do better? We argue that the community has so far made one implicit assumption: that there are no ways to actively control and suppress actual model capabilities (i.e., language model policies) at test time in a manner that is fully reversible and adapted to each user. This assumption is also evident in how large AI firms, such as OpenAI and Anthropic, release their models: every user has access to the same base language model policy, with identical internal weights. This approach is natural: *how could it be otherwise?* After all, alternatives, such as deploying separate models with user-specific parameters, are inefficient and costly, while output-level control provides most guarantees.

Yet this approach comes with a major sacrifice: language model policies always retain the same weights, which encode unwanted or unnecessary knowledge for a given task. In this work, we challenge the premise that we ought to accept this sacrifice.

We argue that an alternative, so far unexplored solution to the deployment-time control problem is to reframe it as a *least-privilege problem*, following the principle of least privilege in computer science. That is, we aim to provide users only with the access they need for their specific tasks and suppress all other knowledge within the language model. **No More, No Less**. We call the extent of such knowledge "privilege." Such deployment-time control should determine how much internal capability a model is allowed to use at inference time, without updating base weights, under adaptive prompts, while remaining reversible and configurable on a per-user basis. Notice that this is a broad class of problems (i.e. this is not constrained to the safety problem used as a motivating example).

Now, we are left with a clear problem: *what does it practically mean to give a user only the required privilege level?* Prima facie, this may seem impossible. Providing less prompt-level information does not remove knowledge from the language model, while deploying a separate model for each user is not a cost-viable solution. We instead propose to look inwards and define *privilege* as the accessible model class. Under this definition, reducing privilege restricts the policy space to a lower-dimensional subset of the original space (i.e., it literally reduces the computations the model can perform). We make this concrete by turning privilege into a *deployment-time control variable* that is enforced *inside* the forward pass by restricting available computations. To do so, we require a language of enforcement at

deployment. We therefore formalize control as a monitor–allocator–enforcer stack (Fig. 1). In this stack, request-time signals (the prompt as well as available meta-data) are mapped by an allocator to a privilege setting, and an enforcer implements this privilege as an operator that affects which internal computations are reachable (while preserving the shape and size of the language model's weights). Stressing once more: under this view, least privilege is a mechanism that *shrinks what the model can do* on a given request, doing so reversibly and without updating the weights.

This framing is deliberately distinct from adaptive-compute methods such as early exit or mixture-of-depths, which vary *how much* computation a request spends while leaving the model's function class intact, so that any behavior the base model admits stays reachable under repeated sampling. Restricting reachable computation instead shrinks the function class itself, which makes the approach complementary to alignment and output-level filtering rather than a replacement: it composes with existing safeguards and turns the privilege granted to each request into an explicit, auditable quantity. To make this concrete (and show this is, in principle, a viable alternative paradigm), we propose a specific instantiation of a least-privilege network, which we call *Nested Least-Privilege Networks* (NLPNs). These consist of a rank-indexed, monotone family of internal linear maps whose reachable subspace is selected by a deployer-chosen privilege setting (Fig. 1). This yields a family of deployed policies that occupy a privilege–utility frontier (Fig. 2). The primary goal of our instantiation is to showcase the viability of the least-privilege paradigm. The remainder of the paper follows this arc: we operationalize the least-privilege problem by defining privilege as reachability and decomposing deployment control (Sec. 3); we formalize least-privilege inference as an allocator objective over a fixed enforcement surface; we instantiate enforcement with NLPNs (Sec. 4); and we evaluate whether the resulting control knob is policy-usable by tracing privilege–utility–overhead trade-offs and demonstrating targeted, selective capability suppression with limited collateral degradation (Sec. 5).

**Contributions**. (1) We identify limitations in existing deployment-focused approaches (Sec. 2); (2) We introduce a new class of models: *least-privilege language models* (Sec. 3); (3) We instantiate an enforcement mechanism with *Nested Least-Privilege Networks* (NLPNs) (Sec. 4); (4) We run extensive experimental evaluations across datasets, models, control policies, and privilege-utility frontiers (Sec. 5).

**Conflict of Interest Disclosure**. The language models we evaluate (Qwen2.5, Qwen3, Llama-3.2, and Pythia) were developed by third parties, and no author is employed by, or has a financial relationship with, any of their developers.

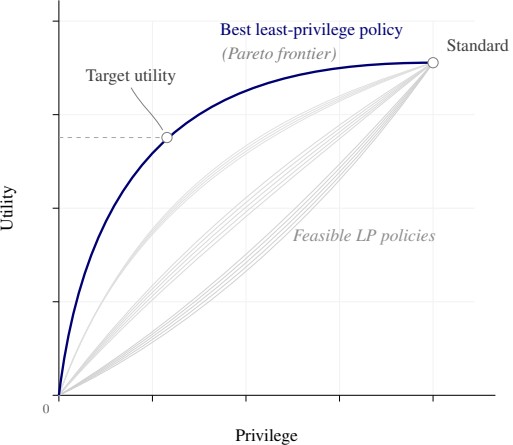

*Figure 2.* **Privilege–utility frontiers induced by inference-time control.** Varying the privilege level $g$ yields a family of deployed policies $\{\pi_{\theta,g}\}$ that trade task utility against a proxy for internal capability (e.g. average rank used). Full-privilege operation occupies the high-capability extreme, while least-privilege policies aim to meet a fixed utility target with minimal average privilege by allocating higher $g$ only to hard or uncertain requests. This frontier is the object optimized by least-privilege inference: minimize privilege subject to a required utility constraint. We show a practical example of this frontier with a language model in Sec. 5.

## 2. Problem Setup and Enforcement Desiderata

### 2.1. Preliminaries

Let $p_\theta(\cdot \mid x)$ denote a fixed pretrained language model with parameters $\theta$. In deployment, the base model is almost never exposed directly: it is mediated by a *deployment stack* (e.g., system prompts, tool routing and constraints, filters, refusal logic, and other wrappers) that induces an effective *deployed policy* $\pi_\theta(\cdot \mid x)$ over outputs for each request $x$.

We study an additional deployer-chosen *inference-time control setting* $g \in \mathcal{G}$ that is enforced inside the forward pass. Concretely, $g$ indexes an enforcement operator $T_g$ applied at inference time, producing effective parameters $\theta(g) := T_g(\theta)$, and thus a controlled family of deployed policies $\pi_{\theta,g}(\cdot \mid x) := \pi_{\theta(g)}(\cdot \mid x)$ with $T_{g_{\max}}(\theta) = \theta$, so that the full setting $g_{\max}$ recovers the original deployed behavior.

We evaluate any fixed setting $g$ on a workload modeled as a distribution over requests $x \sim \mathcal{X}$ together with a bounded task score $r(x, y) \in [0, 1]$ (e.g., accuracy, success, or preference). The induced *utility* at setting $g$ is

$$\mathcal{U}(g) := \mathbb{E}_{x \sim \mathcal{X}} \, \mathbb{E}_{y \sim \pi_{\theta,g}(\cdot \mid x)}[r(x, y)].$$

### 2.2. Least-privilege inference objective

The controlled family $\{\pi_{\theta,g}\}_{g \in \mathcal{G}}$ induces a deployment-time decision problem: for each request $x$, an allocator chooses a setting $g(x)$, which is then implemented by the enforcer

through $T_{g(x)}$. The resulting deployed behavior is therefore request-conditional: $y \sim \pi_{\theta,g(x)}(\cdot \mid x)$. A natural objective is to meet a required level of task performance while choosing the *smallest* setting consistent with that requirement. Concretely, fix a target utility level $u_0 \in [0, 1]$. We define least-privilege inference as the problem of choosing an allocator $\phi$ (equivalently, a mapping $x \mapsto g(x)$) that minimizes the average setting subject to achieving the target utility:

$$\min_{\phi} \ \mathbb{E}_x[g(x)] \quad \text{s.t.} \quad \mathbb{E}_x \, \mathbb{E}_{y \sim \pi_{\theta,g(x)}(\cdot|x)}[r(x,y)] \ \geq \ u_0,$$

where $g(x) = \phi(s(x))$ when the allocator acts on signals $s(x)$. Eq. (2.2) is stated at the level of deployment-time operation: the model parameters $\theta$ are fixed, and the only per-request action is the choice of $g$ from the enforcer's interface.

### 2.3. Desiderata for least-privilege language models

Least-privilege operation is only meaningful if the deployer can reliably *set* such privilege. Given the discussion above, we argue that there are four key desiderata that least-privilege language models ought to satisfy.

> • **D1**. *Can I set privilege precisely and reversibly?* There must exist a single parameterized system that can be run at multiple privilege levels $g \in \mathcal{G}$.
> • **D2**. *Will setting the privilege actually restrict internal reachability?* Privilege must correspond to a change in what internal computations are reachable at inference.
> • **D3**. *When I change privilege, do I get a stable and optimizable trade-off between privilege and utility?*
> • **D4**. *Does privilege affect different tasks and components differently?* Least-privilege allocation should not degrade all behaviors uniformly.

How can we build language models which satisfy the desiderata above? In what follows, we introduce a framework to reason about these challenges (Sec. 3) and propose a concrete instantiation, *Nested Least Privilege Networks* (Sec. 4) that directly address these properties.

## 3. Least-Privilege Language Models

### 3.1. Privilege as reachable internal computation

Hitherto, $g$ has served as an inference-time control setting that indexes a controlled family of deployed policies $\{\pi_{\theta,g}\}_{g \in \mathcal{G}}$ through an enforcer $T_g$. We now give this control a semantic meaning that matches the least-privilege principle: we interpret $g$ as a *privilege setting* that determines how much internal capability the model is allowed to exercise during inference.

**Definition 3.1 (Privilege as reachability)** *For each $g \in \mathcal{G}$, the enforcer $T_g$ induces effective parameters $\theta(g) := T_g(\theta)$ and a deployed policy $\pi_{\theta,g}(\cdot \mid x) := \pi_{\theta(g)}(\cdot \mid x)$, with*

$T_{g_{\max}}(\theta) = \theta$. *We say the mechanism induces an ordered privilege interface if increasing $g$ weakly enlarges the set of reachable forward-pass computations: any internal computation reachable at $g_1$ is also reachable at any $g_2 \geq g_1$ by setting the additional degrees of freedom available at $g_2$ so as to recover the $g_1$ computation.*

By construction, lowering privilege shrinks the set of computations the model can realize, so $\pi_{\theta,g}$ should be interpreted as a *weaker* policy: it may be strictly less capable, in that there exist workloads on which its best-achievable performance is lower. Because lower privilege implies a weaker policy, we therefore generally expect that for two privilege levels $g_1 \leq g_2$, this induces the following policy distribution under the task score:

$$\mathbb{E}_x \, \mathbb{E}_{y \sim \pi_{\theta,g_1}(\cdot|x)}[r(x,y)] \ \leq \ \mathbb{E}_x \, \mathbb{E}_{y \sim \pi_{\theta,g_2}(\cdot|x)}[r(x,y)].$$

### 3.2. Decomposing deployment control

Deployment-time control is easiest to reason about if we separate *what we measure*, *what we decide*, and *what we can actually enforce*. With this in mind, we propose to decompose a deployed system into three layers.

• **Layer 1: Signals (monitoring).** A monitor produces request-time signals $s(x)$ from the input. Broadly, this represents whether we need to change the privilege, either due to security concerns or to give more computational power to resolve a task. These signals can be computed at deployment even when the base weights $\theta$ are fixed.

• **Layer 2: Allocator (decision rule).** An allocator is a decision rule $\phi$ that maps signals to a privilege setting: $g \leftarrow \phi(s(x))$. This is the deployment-time control policy: it decides how much internal capability to grant for the current request, possibly conditioned on user role, risk tier, and uncertainty. In the least-privilege literature outside of language models, it is common to also compute some cost for privilege levels. While we can do that by defining a scalar cost proxy, we avoid it for the purposes of this work.

• **Layer 3: Enforcer (inference-time mechanism).** An enforcer implements the control by applying the inference-time operator $T_g$ *inside the forward pass*, yielding $\theta(g) = T_g(\theta)$ and thus a deployed policy $\pi_{\theta,g}(\cdot \mid x) = \pi_{\theta(g)}(\cdot \mid x)$. This layer defines the deployer's action space: what settings $g$ exist, how they are composed, and what privilege–utility trade-offs they induce.

> 💡 **Takeaway**. A *least-privilege language model* is a deployment object defined by the composition of the three layers. A request $x$ is mapped to signals $s(x)$, the allocator commits to an action $g = \phi(s(x))$, and the enforcer instantiates the corresponding policy with $\theta(g) = T_g(\theta)$, yielding $y \sim \pi_{\theta,g}(\cdot \mid x)$.

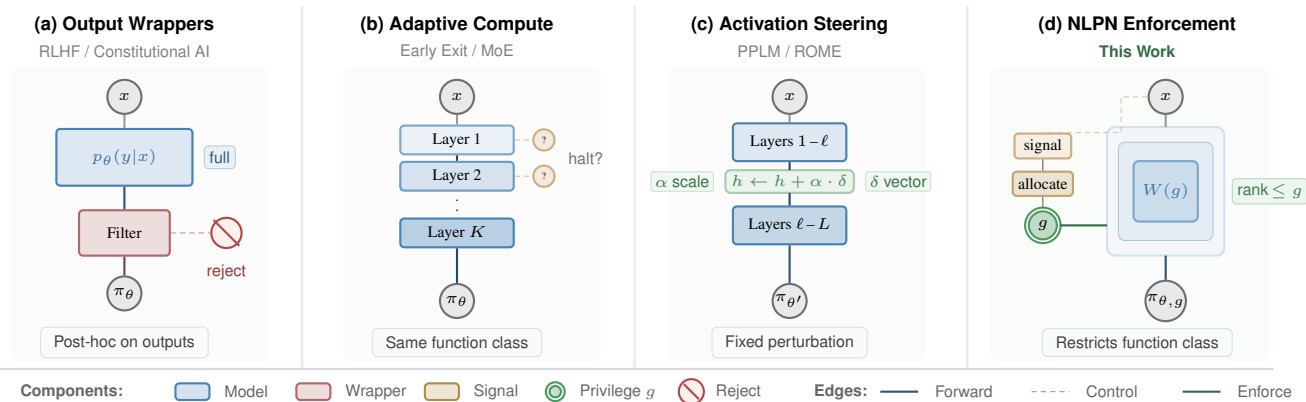

Figure 3. **Taxonomy of deployment-time control mechanisms.** (a) Output wrappers filter after full computation; the underlying capability remains reachable via repeated sampling. (b) Adaptive compute varies depth/budget but preserves the model's function class. (c) Activation steering injects a fixed perturbation at a chosen layer. (d) **Least Privilege Models** (this work): request-time signals determine privilege level $g$, which constrains the rank of internal weight matrices, restricting the reachable function class during inference.

## 4. Nested Least-Privilege Networks

The hardest part of the least-privilege problem is the enforcement mechanism: how can we enforce that computations are in fact suppressed? We provide a first approach to solving this problem.

### 4.1. Enforcing and controlling privilege

**How to enforce privilege?** We make the insight that we can enforce privilege by selectively reducing appropriate *rank* of the parameters, and learning a function to switch between relevant ranks at test time (Rauba & van der Schaar, 2025). With this, we implement $T_g$ by replacing selected linear maps in the transformer with *Nested Least-Privilege Networks* (NLPNs). Consider a linear map $W \in \mathbb{R}^{d_{\text{out}} \times d_{\text{in}}}$. An NLPN re-parameterizes $W$ by a fixed maximum rank $r_{\max}$ and factors, where $W \approx BA$, where $A \in \mathbb{R}^{r_{\max} \times d_{\text{in}}}$, $B \in \mathbb{R}^{d_{\text{out}} \times r_{\max}}$. For privilege $g \in \{0, 1, \dots, r_{\max}\}$, we define the prefix submatrices $A_g \triangleq A_{(1:g,:)}$, and $B_g \triangleq B_{(:,1:g)}$, and the effective weight

$$W(g) \triangleq B_g A_g = \sum_{i=1}^{g} B_{(:,i)} A_{(i,:)}.$$

The enforcer $T_g$ applies this substitution (simultaneously across all NLPN-modified layers), producing effective parameters $\theta(g) = T_g(\theta)$ and the deployed policy $\pi_{\theta,g}$.

**How to control privilege at test-time?** NLPNs induce an ordered privilege interface. For any $g < r_{\max}$, $\text{Im}(W(g)) \subseteq \text{Im}(W(g+1))$, because $W(g)x$ lies in the span of the first $g$ columns of $B$, and $W(g+1)x$ lies in the span of the first $g+1$ columns of $B$, which contains the former. As $T_g$ applies this nesting consistently across layers, increasing $g$ weakly enlarges the set of reachable forward-pass computations, while decreasing $g$ restricts it.

### 4.2. Post-hoc training: making $g$ a stable deployment knob

We still require an application procedure to LLMs as well as a loss function that would enable trading off different privilege levels. In particular, optimizing only at $g = r_{\max}$ and truncating at test time can produce fragile low-privilege behavior. We therefore fine-tune NLPN factors so that both a high-privilege policy and a sampled low-privilege policy are directly optimized.

**Sampling behavior.** At training, at each step we sample a *variant* privilege $g$ uniformly from $\{0, 1, \dots, r_{\max} - 1\}$ and always include the *anchor* privilege $r_{\max}$. This trains the hierarchy without evaluating all privileges per step.

**Computing the loss function.** Let $\mathcal{L}_{\text{CE}}(g)$ denote the cross-entropy loss computed under $\pi_{\theta,g}$. We learn a rank-specific log-variance $s_g$ and use the standard uncertainty-weighted surrogate for the two evaluated privileges:

$$\mathcal{L}_{\text{total}} = \left( \exp(-s_{r_{\max}}) \mathcal{L}_{\text{CE}}(r_{\max}) + s_{r_{\max}} \right) \\ + \left( \exp(-s_g) \mathcal{L}_{\text{CE}}(g) + s_g \right).$$

This is an uncertainty-based weighted surrogate equivalent objective adapted from the multi-task learning literature (Kendall et al., 2018). The exponential factors scale gradient contributions from each evaluated privilege, while the $+ s$ terms prevent privilege collapse. The full implementation is given in Alg. 1.

**Summary.** We replace each transformer linear layer with a product of two lower-rank matrices, initialized via SVD. All (or selected) LLM layers / blocks are reparameterized, and we fine-tune with the given loss function. This enables flexible rank truncation at inference time.

**Algorithm 1** Post-hoc NLPN training with anchor privilege $r_{\max}$ and sampled variant privilege $g$

---

**Require:** Pretrained parameters $\theta$; maximum rank $r_{\max}$; optimizer Opt
**Require:** Log-variances $\{s_g\}_{g \in \{0,\ldots,r_{\max}\}}$ (initialized to 0)
1: Replace selected $W$ by NLPN factors $(A, B)$ with $A \in \mathbb{R}^{r_{\max} \times d_{\text{in}}}$, $B \in \mathbb{R}^{d_{\text{out}} \times r_{\max}}$, initialized so that $B_{(:,1:r_{\max})} A_{(1:r_{\max},:)} \approx W$
2: **for** each training step with minibatch $(x, y)$ **do**
3:     Sample $g \sim \text{Uniform}(\{0, 1, \ldots, r_{\max} - 1\})$
4:     Compute $\mathcal{L}_{\text{CE}}(r_{\max})$ using $\pi_{\theta, r_{\max}}$
5:     Compute $\mathcal{L}_{\text{CE}}(g)$ using $\pi_{\theta, g}$
6:     Form $\mathcal{L}_{\text{total}}$ via Eq. (4.2)
7:     Opt.zero_grad(); backprop through NLPN factors and $\{s_g\}$; Opt.step()
8: **end for**

---

## 5. Illustrative Studies

Before treating the enforcement surface as a control object, we verify the interface is well-formed. For this check we fine-tune the NLPN factors for next-token prediction on a general-domain corpus, MiniPile (Kaddour, 2023); without such nested training, naïve rank truncation collapses (Fig. 4), whereas NLPN tuning makes perplexity rise only gradually across ranks. With this precondition, we turn to our research questions (**RQs**).

### 5.1. RQ1: Does reducing privilege yield a utility degradation at inference time?

This question matters since least privilege is operational if we can in fact use $g$ at test-time and it induces a smooth degradation in utility (Desiderata 1 & 3).

We construct algorithmic tasks where difficulty can be scaled, enabling clear diagnosis of whether rank reduction yields a stable control surface. We train and evaluate multiple LLM models on three tasks: Balanced Brackets, Length Comparison, and Contains Substring-sampling $N$ examples per difficulty level (See Appendix C) and evaluating across multiple rank reductions.

**Analysis**. Fig. 5 shows that decreasing privilege reduces utility in a monotone manner. That is, easy instances maintain near-saturated performance at moderate ranks, while harder instances show sharper degradation. This is the *differential sensitivity* property discussed in Sec. 2.3.

> **Insight 1**. Decreasing privilege monotonically reduces the performance on hardest tasks. Similarly, easy problems remain solvable at low privilege.

### 5.2. RQ2: Can we compare the results of privilege allocation policies for a given utility target?

We wish to understand whether there are some policies which are *better* (i.e. have strictly higher utility for a given rank). We therefore implement 5 privilege allocation policies. **Policies tested**: *Full-privilege* and *Min Rank* always sets the max and min privilege, respectively; *Static LP* selects a single $g$ calibrated from validation data; *Progressive escalation* increases $g$ step-wise based on uncertainty.

**Analysis**. Fig. 7 shows that, at each required accuracy target, we can compare allocators by the average rank they consume. Static full-privilege sits at the high-privilege extreme; static min-rank often fails at higher targets. Static LP chooses a single operating point; progressive escalation achieves comparable targets with less average privilege by allocating high rank only to hard/uncertain instances.

> **Insight 2**. Policies trade off privilege and utility and we can obtain Pareto-optimal policies.

### 5.3. RQ3: Can we evaluate policies by privilege, utility, and overhead?

Some policies may be preferred in different circumstances; here we evaluate the trade-offs between utility, privilege, and overhead.

**Analysis**. Fig. 6 shows that different policies have clear trade-offs: either providing too much privilege (e.g. max-rank) or failing to obtain the required utility (e.g. min-rank). Some allocation policies, e.g. progressive incremental policies, are both dynamic and give the least privilege. When the target is low (80%), many policies converge to low rank with minimal overhead; when the target rises (90–95%), allocators diverge. Static LP must choose a single rank that satisfies the worst of the distribution, increasing privilege even on easy cases. Progressive escalation is a conditional strategy: it pays overhead on hard cases but reduces average rank by keeping easy cases at low privilege. The "jump" variant shows a second axis: allocators can trade privilege for fewer passes by jumping to $g_{\max}$ only when needed (more evals in App. C.4). **In practice**: use static LP when latency or throughput dominates; use progressive (incremental) when minimizing average privilege matters, accepting modest extra latency (in our runs, $\approx$2–3$\times$ passes); and use progressive (jump) for the middle ground, trading some privilege efficiency for lower overhead ($\approx$1.3–1.5$\times$).

> **Insight 3**. Some policies, especially dynamic policies which adjust their privilege based on content, achieve required utility targets while minimizing privilege with minimal overhead.

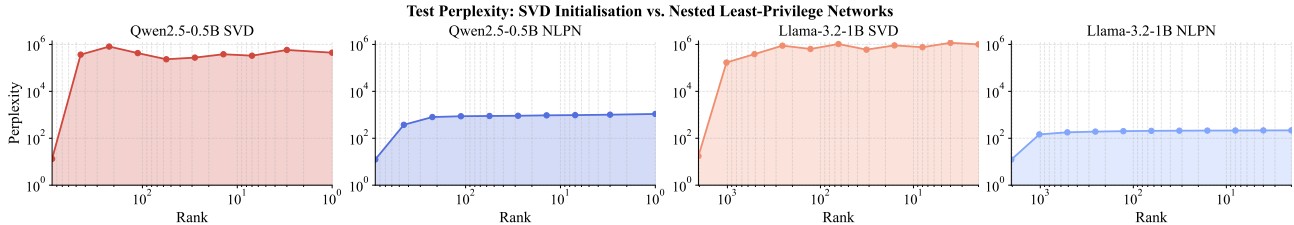

*Figure 4.* **Perplexity vs. MLP rank: SVD vs. NLPN.** Qwen2.5-0.5B and Llama-3.2-1B under (1) SVD initialisation and (2) NLPN tuning on MiniPile. SVD shows catastrophic degradation; with NLPN tuning, perplexity rises only gradually across ranks via nested subspace training. MLP projections are intervened uniformly across all transformer blocks.

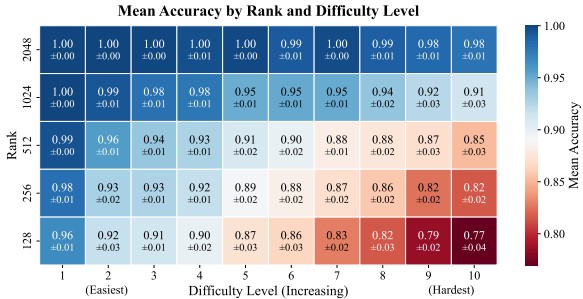

*Figure 5.* **Performance on Pythia-1B by rank (privilege) and difficulty level**. Accuracy degrades as rank is reduced, with larger drops at higher difficulty, performed on the balanced brackets task. This property is desirable: this means privilege can be selectively tuned to the difficulty of the task.

### 5.4. RQ4: Can we suppress specific architectural blocks to reduce required privilege on some tasks while preserving utility on others?

To suppress required capabilities, it would be useful to identify block-level features, such that the rank reduction within a given block corresponds to decreases in some required areas but not others. To investigate this, we replace feedforward modules in pre-trained causal language models with NLPN layers and fine-tune on MiniPile for around 1B tokens (taking 3-4 hours on an NVIDIA GeForce RTX 4090).

**Analysis**. Fig. 8 demonstrates the case for fine-grained privilege. Specifically, we suppress only *selected* blocks to observe how performance changes at different tasks. We hope to find differential sensitivity, i.e. some blocks suppressing some knowledge; but not others. We find that many block/projection interventions have negligible effect, while a small subset produces significant subject-specific degradation. This suggests that a deployer can expose a structured control vector (instead of a global rank) allowing allocators to reduce privilege.

> **Insight 4**. There are block-level localizations that, when suppressed, reduce privilege on a subset of tasks whilst maintaining utility on others.

### 5.5. RQ5: Can we directly *optimize* a model to reduce privilege on custom tasks?

We now care about directly optimizing a model such that there is reduced privilege on custom tasks that the deployer of a language model chooses. This is important for selecting what privilege to grant from the perspective of the deployer.

**Analysis**. We perform a lightweight optimization procedure and see that we can directly suppress chosen subjects (Fig. 9). We find we can consistently limit privilege to some knowledge class (with examples of suppressing knowledge in chemistry, *or*, chemistry and biology) with no degradation in other performance metrics. We further find that there are many possible arrangements of blocks which could result in such suppression. In fact, we can even enumerate all such configurations and evaluate their effect on the suppressed class. These results give overwhelming evidence that selective privilege allocation is feasible.

> **Insight 5**. We show how to perform lightweight optimization to obtain least-privilege on specific tasks while ensuring high utility in other tasks in language models.

### 5.6. Further least-privilege studies

We wish to understand when does least-privilege work, why, and whether it can be used as a novel deployment paradigm for language models. To do so, we perform extensive evaluation across other setups (full results in Appendix C - F).

**(a)** We construct controlled-difficulty algorithmic workloads for least-privilege allocation (Tab. 1). **(b)** We report aggregate accuracy across model–task–rank settings (Tab. 2). **(c)** We show rank reduction disproportionately hurts hard instances vs. easy ones, enabling conditional privilege allocation (Fig. 11). **(d)** We quantify utility retention vs. privilege reduction for multiple allocators relative to full privilege (Tab. 3). **(e)** We summarize the utility–privilege–overhead trade-off space across allocators and targets (Tab. 4). **(f)** On *Contains Substring*, we show uncertainty-based escalation reduces average privilege while jump escalation trades priv-

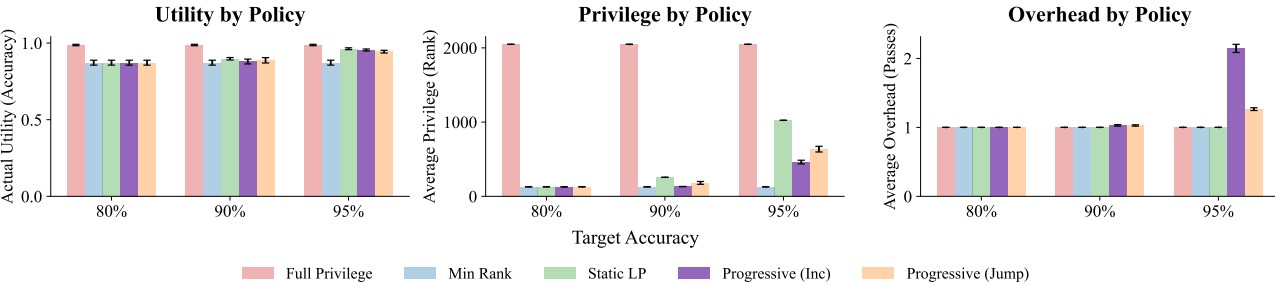

*Figure 6.* **Utility-Privilege trade-offs for five policies on Pythia-1B**: Policy comparison across target accuracies (80%, 90%, 95%). At low targets, policies converge to minimal privilege; at higher targets, allocators diverge significantly. Static LP must over-allocate privilege to satisfy worst-case instances; progressive escalation adapts rank to instance difficulty, reducing average privilege at the cost of additional inference passes. The jump variant demonstrates the privilege - overhead trade-off by escalating directly to maximum rank when needed.

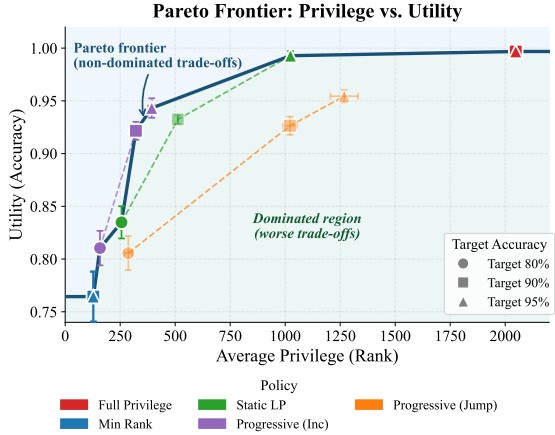

*Figure 7.* **Utility-Privilege Pareto Frontier on Pythia-1B**: privilege–utility frontier induced by several simple allocators at target accuracies (80%, 90%, 95%) on the balanced-brackets task. Progressive escalation traces lower average rank than static full privilege at fixed targets, while trading off additional passes.

ilege for fewer passes (Fig. 12). **(g)** We show single-MLP rank control yields smooth, projection-dependent MMLU degradation surfaces (Figs. 13a, 13b). **(h)** We find sparse, localized block–module sensitivity under BH-FDR masking, supporting granular control (Benjamini & Hochberg, 1995) (Figs. 14a, 14b) which allows demonstrating selective subject suppression with sparse interventions and limited collateral damage (Fig. 9). **(i)** We show rank reduction corresponds to true capacity suppression rather than output refusal, as probe recoverability collapses at low rank (Fig. 15).

### 5.7. Naturalistic tasks and larger models

Our main studies use synthetic, difficulty-controlled tasks on sub-1B models, so we ask whether the privilege–utility structure survives a more naturalistic task and larger models. We add a named-entity-recognition (NER) verification task built from real English text, binned into the same ten difficulty levels and evaluated under the identical rank-controlled protocol: on Pythia-1B at the same five ranks as our synthetic study, the RQ1 pattern reproduces almost exactly (Fig. 10; cf. Fig. 5), with monotone degradation in rank and the hardest instances degrading first. The same qualitative behavior recurs across five models spanning 0.5–4B; we claim only this reproduction here, deferring the per-model scaling magnitudes—which are confounded by training budget—to App. G, which reports the full per-model accuracy tables together with the corresponding rank–difficulty heatmaps and degradation curves.

> **Insight 6.** The privilege–utility structure is not an artifact of synthetic tasks or sub-1B models: the RQ1 degradation pattern reproduces on a naturalistic NER task and recurs across models from 0.5B to 4B parameters.

## 6. Discussion

**On related work**. Prior work largely controls LMs either *normatively* by changing parameters via post-training alignment and safety fine-tuning, e.g. instruction/RLHF and preference methods (Ouyang et al., 2022; Bai et al., 2022a; Rafailov et al., 2023) and constitutional or safety objectives (Bai et al., 2022b; Choi et al., 2024), or *behaviorally* at deployment via prompting and wrapper-level filtering/repair (Zheng et al., 2024; Han et al., 2025), which can suppress outputs without removing the underlying capability from the base model. A separate thread studies adaptive inference and decoding for efficiency, including early-exit and dynamic-depth ideas (Zhou et al., 2020; Bae et al., 2023) and decoding accelerations (Xia et al., 2024). Unrelated to least-privilege but closest to our interface are ordered-capacity architectures such as slimmable and once-for-all networks (Yu et al., 2018; Yu & Huang, 2019; Cai et al., 2019), which expose families of subnetworks indexed by width or depth. While some might be subsumed as alter-

NLPN Qwen2.5-0.5B BH-FDR Masked MMLU Task ΔAccuracy with Down Projection Rank Reduction

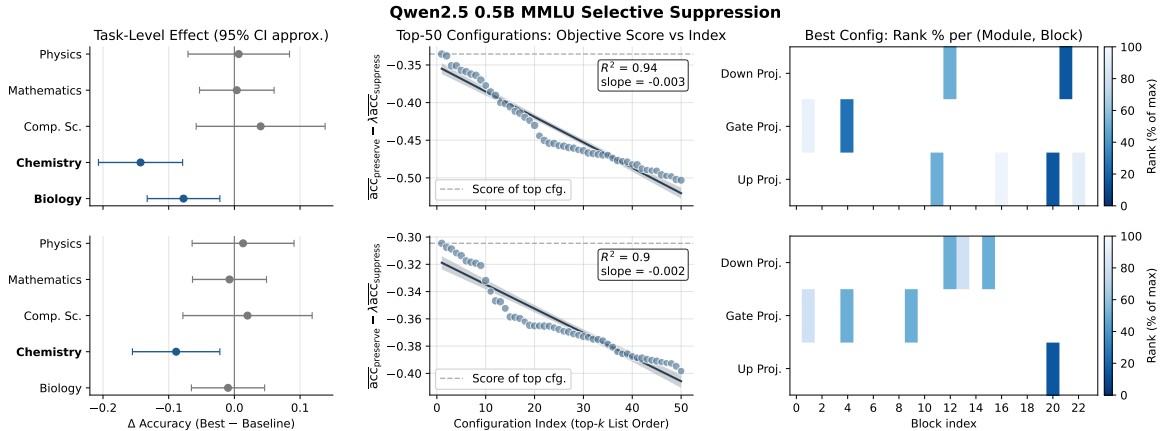

*Figure 8.* **Down proj. block-rank sensitivity in NLPN-Qwen2.5-0.5B.** Δ accuracy vs. full-rank baseline by task. Non-significant changes masked grey (Benjamini & Hochberg, 1995). Dot size increases *with the number of consecutive ranks at which a significant effect is observed for a given* (`task, block`). Sparse patterns enable granular privilege control. More examples shown in App. E.

*Figure 9.* **Selective capability suppression. Top:** Biology + Chemistry suppression. **Bottom:** Chemistry suppression. Targets are degraded while preserving others, balancing suppression efficacy against collateral damage. **Left:** Per-task accuracy changes for the best configuration. **Middle:** Score across optimisation candidates, obtained with beam search optimisation described in App. E.2. **Right:** Spatial structure of the best configuration—heatmap shows rank percentages for each MLP in the model. Sparse intervention patterns demonstrate that selective control requires only localised reductions.

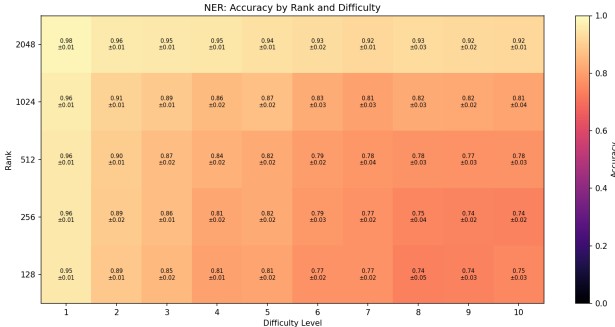

*Figure 10.* **Naturalistic NER verification on Pythia-1B by rank (privilege) and difficulty level.** The same model and rank schedule as the synthetic Balanced Brackets heatmap (cf. Fig. 5), now on a named-entity-recognition verification task built from real text. Accuracy degrades monotonically as rank is reduced, with larger drops at higher difficulty: the RQ1 differential-sensitivity pattern reproduces on naturalistic input.

native enforcement mechanisms (an encouraging research direction), NLPNs differ by providing a post-hoc, shape-preserving, rank-indexed enforcement interface that sup-

presses reachable internal computation, composes with the monitor–allocator–enforcer stack for per-request privilege ceilings, and can be optimized for selective task-dependent degradation. Finally, inference-time intervention work modifies forward-pass activations or weights (steering and editing/erasure) (Turner et al., 2024; Ravfogel et al., 2022a), typically via fixed perturbations or targeted edits rather than an ordered, reversible privilege interface.

**Least-privilege as a way to move beyond alignment**. The primary purpose of this paper is to re-consider what we deploy as language models. We challenge the common notion that existing language modeling interfaces must always expose full capabilities to each user. We establish a paradigm of language modeling called *least-privilege language models* as a novel way to expose only certain capabilities to users via the same interface by assigning custom privilege based on the request; and exemplify our work with NLPNs. Above all, we view this work as defending a new deployment paradigm that encourages research into providing custom access within a single language modeling interface, insofar as fundamental limits permit (Stadler et al., 2024).

## Acknowledgements

This work was supported by Azure sponsorship credits granted by Microsoft's AI for Good Research Lab. The authors would like to further thank Jev Gamper and Augustas Macijauskas for helping to form the group of authors who wrote this work.

## Impact Statement

This work proposes deployment-time interface for constraining which internal computations a language model can execute, enabling per-request "capability minimization" without changing the underlying base weights. Such interfaces could improve safety and governance by reducing unnecessary capability exposure on benign requests and by making access decisions explicit and auditable. However, they are not standalone guarantees for safety (and can be used for areas outside of safety, as showcased in our experiments). Suppressed capabilities may be recoverable under adaptive prompting or downstream fine-tuning, and effectiveness depends on the allocator signals, evaluation protocol, among others. This work lays the ground work for least-privilege inference and we therefore do not yet know the extent to which recoverability or suppression are feasible within the larger machine learning domain. While our work has demonstrated that we can selectively suppress and regulate privilege while maintaining utility, the impacts or theoretical guarantees for when this holds require future research. Furthermore, the same least-privilege mechanisms could be misused to implement opaque censorship, discriminatory access tiers, or selective information suppression, so any real deployment should pair least-privilege controls with transparency, monitoring, and red-team evaluation, and should be assessed for distributional impacts across user groups. We position this contribution as infrastructure for studying measurable privilege–utility–overhead trade-offs and for enabling more rigorous, testable safety claims. Lastly, we see the potential for least-privilege to be adopted as a novel language modeling interface across many industrial applications and therefore believe there are many downstream social impacts of this work.

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

# A. Extended related work

## A.1. Broader overview of related work

**Training-time alignment**. While our work does not directly target training-time alignment, it is relevant inasmuch as training-time alignment and least-privilege language modeling, as a paradigm, to enable us to change the deployed behaviour of language models. Training-time alignment focuses on reshaping output distributions or filtering what outputs are emitted. RLHF, beginning with preference-based reward signals (Christiano et al., 2017; Ziegler et al., 2019; Stiennon et al., 2020). Such approaches scale into instruction-following systems (Ouyang et al., 2022); and were extended to building "helpful/harmful" objectives (Askell et al., 2021; Bai et al., 2022b). Beyond RLHF or other related strategies (Rafailov et al., 2023), others have explored safety-specific fine-tuning objectives (Choi et al., 2024); this, in turn, motivates analyses of which parameters govern safety compliance (Qi et al., 2023; Li et al., 2024b; Huang et al., 2024). While such alignment is useful for training-time, this is orthogonal to our line of work which focuses on re-thinking how we deploy language models. That is, our work argues that we can deploy language models to contain least-privilege for each custom user; which can be done together with training-time alignment.

**Adaptive computation**. A category of tangentially related work is on adaptive inference, i.e. dynamically adjusting a model's computation per input. Such methods aim to improve efficiency or latency; and therefore contrast with our goal to enable users to obtain the least amount of privilege, e.g. as information contained in the forward pass. One class of methods in adaptive computation is called Early-exit Transformers which attach internal classifiers at multiple layers (Zhou et al., 2020; Xin et al., 2020; Liu et al., 2020). This has been applied to autoregressive tasks too: we can now train language models with a policy to exit early if the subsequent tokens are predictable (Bae et al., 2023). A large class of methods emerged as *mixture-of-depth* approaches (Raposo et al., 2024), or recurrent (universal transformer-based) variants (Csordás et al., 2024). Other approaches achieve significant speed-ups for GPT-style models without changing models, such as using speculative decoding (Xia et al., 2024). Other adaptive computation-time mechanisms have been studied throughout the years, including, famously, dynamic halting (Graves, 2016) which is employed by universal transformers (Dehghani et al., 2018). Other approaches might include selecting the cheapest language policy for a given task or delegating it to a human (Fanconi & van der Schaar) or other router-based mechanisms (Mohammadshahi et al., 2024). Importantly, while all these methods vary the amount or path of computation, they do not restrict the function class of the model. An early-exit or MoE-accelerated model can still, in principle, generate any output that the full model could, it simply might not choose to use all its capacity on every request.

Ordered-capacity architectures such as slimmable networks (Yu et al., 2018; Yu & Huang, 2019) and once-for-all models (Cai et al., 2019) expose a family of subnetworks indexed by width/depth, primarily for efficiency and device scaling. They are used to enable dynamic channel dropping to instantiate sub-networks with varying widths. Such methods could potentially be adapted for least-privilege language modeling and be subsumed under our work. That said, NLPNs are similar in exposing an ordered interface, but differ in (i) enabling post-hoc application to pre-trained language models; (ii) have shape-preserving properties for pretrained transformers, (iii) offer a mathematical mechanism for suppressing reachable computational spaces; (iv) explicitly separating monitor–allocator–enforcer to support governance and per-request privilege ceilings, and (v) can be optimized to reduce utility in some areas but not others. This work should be interpreted as employing ordered-capacity ideas into LLM deployment via access-control rather than proposing a new efficiency method.

**Inference-time activation steering and model editing**. A growing body of work attempts to control or modify model behavior inside the forward pass. Early models introduced activation steering. Some have showcased how to add additive vectors to internal states to elicit or change capabilities (Turner et al., 2023), and solving constraint satisfaction tasks (Turner et al., 2024). This idea has been applied in various areas, e.g. for multiple hallucination categories (Wang et al., 2025), chain-of-thought compression (Azizi et al., 2025), or the area of theorem proving (**?**). In a similar vein, model editing focuses on modeling the weights post-hoc to add, correct, or remove specific knowledge behaviors. For instance, ROME locates facts in the model's MLP layers (Meng et al., 2022), with other approaches editing language model-based knowledge graph embeddings (Cheng et al., 2024) or directly erasing concepts (Ravfogel et al., 2022a;b; Belrose et al., 2023).

**Other related work**. There has been some initial discussion on using least-privilege in the context of machine learning, with two examples being audio models (He et al., 2025) and tool calling for agents (Zhu et al., 2025).

## A.2. Relating to normative and epistemic methods

Another way to view the related work is by employing a categorization of normative and epistemic methods. While our work is not safety per se, we think it might be useful to position it relative to what exists today via this view. We call a method *normative* if it aims to shape what the model *should* do—by changing training data, objectives, or policies so that certain behaviors are preferred and others are discouraged. We call a method *epistemic* if it aims to measure what the model *does* or *can do*—by evaluating behavior (Rauba et al., 2025), eliciting latent capabilities (Donoway et al.), analyzing internal representations (Bullinaria, 2024), understanding where model failure modes are (Rauba et al., 2024) or studying general knowledge capabilities of LLMs (Yildirim & Paul, 2024). Normative methods include pre-training data interventions and post-training alignment procedures (e.g., supervised fine-tuning and preference optimization), while epistemic methods include evaluation, red teaming, elicitation, and mechanistic analyses (Zou et al., 2023; Jiang et al., 2024; Ganguli et al., 2022; Li et al., 2024a).

During pre-training, normative control is available through data and objective design, and epistemic work is used to track capabilities and failures during training. During post-training, normative control is strongest—alignment procedures can directly push behavior toward desired outcomes—and epistemic work is used to audit, stress-test, and diagnose failure modes before release (Jiang et al., 2024; Zou et al., 2023). At deployment, however, the situation changes: epistemic monitoring remains available and can supply signals to a controller, but normative changes to the underlying model are slow or infeasible. As a result, the lifecycle naturally separates phases where we can change $\theta$ from the phase where we must act with $\theta$ fixed, and deployment control becomes a question of what inference-time levers remain.

We view our work as *structural* in the sense that we do not suggest *what to respond* (normative) or *what the model does* (epistemic), but *what is reachable at test time*.

# B. Least Privilege Language Models (cont.)

**Note**: there is no reason to believe a single rank corresponds to a particular semantic capability; whether a model can be optimized to reduce the rank so that it corresponds to some capabilities is an empirical research question (which we demonstrate in Sec. 5.5). To stress this once more: we make no assumptions that there exists a one-to-one mapping between rank and any specific capability. We define privilege as an operational proxy that can limit the reachable computations by projecting a part of the weight matrices into the null space. The mechanism of how specific capabilities are suppressed is a matter of optimization procedures which we demonstrate empirically.

We also note that because computation is not necessarily localized, it could be the case that suppressing the rank at a specific layer does not entirely remove capabilities. That said, our experiments provide extensive support that we can reliably reduce privilege in some tasks while maintaining high utility in other tasks.

## B.1. On allocator signals

We have established that the least-privilege language modeling paradigm can take into account signals $s(x)$ to decide how much privilege to give to a user. Such signals can be metadata, such as information on the user location, surfing activity, behavior online, etc. While this framework is suitable in general, we wish to use only the available data at test time within the language model and therefore define the signal as $s(x) = x$ so that the allocator uses the prompt information alone to decide on the privilege level. We show in Sec. 5.5 that this can be optimized via different policies to obtain utility-privilege frontiers; and would be enhanced with additional signals.

## B.2. Why do we need privilege allocation?

**A one-paragraph justification**. At the risk of stating the obvious, privilege allocation is extremely important. Privilege allocation stems from a problem in deployment setups, whereby a single deployed language model is typically run at a fixed internal capability level chosen to satisfy the hardest legitimate requests, even though most requests do not require that level of computation. This is important because benign queries are routinely processed with more internal capability than they need, and any failure of wrapper-level safeguards applies to a maximally capable forward pass. Wrapper-based controls operate by constraining emitted behavior, but they do not alter which internal representations or reasoning pathways are reachable during inference; the same computation is executed regardless of request context, authorization, or risk. Privilege allocation makes internal capability conditional at inference time, allowing the system to default to minimal reachable

computation and to escalate only when signals indicate legitimate need, thereby reducing unnecessary capability exposure without changing the underlying model weights.

**Is privilege allocation, then, not just compute optimal performance?** No. Compute-optimal allocation optimizes how efficiently a fixed computation is approximated, for example by early exiting, skipping layers, or reducing precision, while leaving the model's underlying function class unchanged. These mechanisms reduce average cost but do not restrict which internal computations can in principle be executed: given sufficient retries, longer contexts, or repeated sampling, the same behaviors remain reachable. Privilege allocation differs in kind rather than degree: it restricts the set of internal computations that can be reached at all unless additional privilege is explicitly granted. As a consequence, repeated querying or prompt search cannot recover suppressed capabilities without escalation, making privilege allocation an access-control mechanism over internal computation rather than a runtime optimization over an unchanged model.

### B.3. The challenges with existing approaches to privilege allocation

There are at least three key issues with privilege allocation as it is done today, which we review here.

▶ **First, existing approaches limit knowledge at the wrapper level**. Deployment-time approaches implement *privilege allocation* only at the wrapper level: they modify the deployed behavior distribution $\pi_\theta(\cdot \mid x)$ while leaving the underlying model $p_\theta(y \mid x)$ unchanged. Wrapper-only privilege allocation changes the deployed behavior $\pi_\theta(\cdot \mid x)$ while leaving the underlying model $p_\theta(\cdot \mid x)$ unchanged, so there is no implication from "good behavior under $\pi_\theta$" to "the capability is absent from $p_\theta$." Concretely, for each fixed prompt $x$, $\pi_\theta(\cdot \mid x)$ is obtained by post-processing samples from $p_\theta(\cdot \mid x)$ (e.g., filtering/refusal/resampling), so the set of outputs that can occur under deployment is contained in the set of outputs the base model can produce (equivalently, $\mathrm{supp}(\pi_\theta(\cdot \mid x)) \subseteq \mathrm{supp}(p_\theta(\cdot \mid x))$), but the reverse containment need not hold; thus reducing mass on an undesirable behavior at the wrapper level does not remove that behavior from the base distribution. In expectation form, even if the wrapper achieves a small average undesirable score on the workload,

$$\mathbb{E}_{x \sim \mathcal{X}} \, \mathbb{E}_{y \sim \pi_\theta(\cdot|x)}[r(x,y)] \approx 0,$$

this does *not* imply that $\mathbb{E}_{y \sim p_\theta(\cdot|x)}[r(x,y)]$ is small uniformly over prompts, and in open-ended prompt spaces it is plausible (and empirically common) that there exists some $x$ for which $\mathbb{E}_{y \sim p_\theta(\cdot|x)}[r(x,y)] > 0$. Let us consider one example why this might be an issue. It is well-known that you can extract specific parts of the language model's knowledge with specific answers by repeatedly querying the model (Carlini et al., 2022) with some work explicitly quantifying how many samples an adversary would need to generate before a target sequence becomes extractable with a given probability under non-greedy sampling (Hayes et al., 2024). Let us consider one possible approach to addressing models by jailbreaking their safeguards is repeated sampling: even if an undesirable behavior is rare on any single draw, an attacker can query until it appears. Concretely, fix a prompt $x$ and draw $n$ independent outputs $y_1, \ldots, y_n \sim p_\theta(\cdot \mid x)$. To make the event structure explicit, in this paragraph we interpret $r(x,y) \in \{0,1\}$ as an indicator for the behavior of interest (so $r(x,y) = 1$ means the behavior occurs). Then the probability that *at least one* of the $n$ samples exhibits the behavior is

$$\Pr\left[\max_{i \in \{1,\ldots,n\}} R(x,y_i) = 1\right] = 1 - \Pr\left[\forall i, \ R(x,y_i) = 0\right]$$

Therefore, if the base model assigns a nonzero probability to the behavior for a given prompt $x$ (i.e. $\mathbb{E}_{y \sim p_\theta(\cdot|x)}[r(x,y)] > 0$), the probability of observing it increases monotonically with the number of samples $n$; and is then equal to $1 - \left(1 - \mathbb{E}_{y \sim p_\theta(\cdot|x)}[r(x,y)]\right)^n$. This matters because wrapper-level suppression that drives $\mathbb{E}_{x \sim \mathcal{X}}\mathbb{E}_{y \sim \pi_\theta(\cdot|x)}[r(x,y)]$ low does not remove the behavior from $p_\theta(\cdot \mid x)$, and any residual mass can be amplified by repeated querying. In practice this amplification is limited by the difficulty of finding prompts $x$ with appreciable base probability since there is an underlying "cat-and-mouse dynamic", whereby top model providers find effective jailbreaks and immediately patch them up.

▶ **Second, wrapper-level control does not naturally support *differential capability* across users**. Concretely, the deployed system $\pi_\theta(\cdot \mid x)$ is typically a single policy that must serve heterogeneous access requirements, so the underlying conditional distribution $p_\theta(\cdot \mid x)$ (and thus the internal computations used to produce logits) is effectively shared across individuals, with access distinctions implemented only by post-processing through the deployment stack. In many settings, however, capability should be *conditional* on context and authorization: for the same request $x$ (or the same task family), we may want high utility for one class of users while requiring that the corresponding behavior be unattainable for another class, e.g. granting domain-specific knowledge about sensitive chemical hazards to appropriately cleared researchers while

denying it to the general public. With wrapper-only allocation, this distinction is expressed as a constraint on the *outputs* admitted by $\pi_\theta(\cdot \mid x)$ rather than as a change in the reachable computation of $p_\theta(\cdot \mid x)$, meaning the base model continues to represent and potentially express the capability, and the system's separation of access levels is mediated only by the wrapper rather than by an internal privilege setting.

▶ **Third, existing approaches do not support composability under changing deployment constraints**. In practice, deployed systems accumulate heterogeneous and evolving requirements—risk thresholds, content categories, user roles, and operational policies—that must be enforced jointly at inference time. Wrapper-level control implements these requirements by restricting or reshaping the deployed behavior distribution $\pi_\theta(\cdot \mid x)$ through a growing stack of prompts, filters, and decision rules. While each wrapper may be well-motivated in isolation, their composition does not induce a structured or ordered family of deployed policies: adding or tightening a constraint can change behavior in non-monotone and hard-to-predict ways, and there is no canonical notion of "more" or "less" capability across wrapper configurations. As a result, adapting the system to new deployment requirements typically requires ad hoc retuning rather than principled reallocation. This motivates inference-time control mechanisms that expose a stable, composable action space—such as a family $\{\pi_{\theta,g}\}$ indexed by privilege—in which multiple constraints can be expressed and combined by operating on a shared internal control surface rather than by layering additional wrappers.

### B.4. On the limitations of this work

Several limitations remain. First, a privilege interface is not itself a safety solution: capabilities can often be recovered by adaptation (Patil et al., 2023; Lynch et al., 2024), and threat models remain essential (Zou et al., 2023; Jiang et al., 2024). Second, privilege metrics (e.g. average rank) are proxies for reachable computation; connecting them to stronger notions of capability classes is an open direction. Third, the allocator design space is large: monitors and decision rules can incorporate richer signals and interact with tool-use policies and wrappers. Our stance is that these are separable problems: the enforcer should provide a predictable action space; the allocator can then be improved without changing the model weights. Finally, the least-privilege framing suggests a systems-style research agenda: design privilege interfaces that compose (multiple knobs), calibrate uncertainty measures to capability, and integrate enforcement with existing governance stacks that already include monitoring and wrapper constraints.

## C. Algorithmic Tasks: Detailed Performance Analysis

Our implementation, experimental code, and detailed results are publicly available at `https://github.com/pauliusrauba/least-privilege-language-models`.

This appendix provides results from our evaluation of least-privilege inference on three algorithmic reasoning tasks: *Balanced Brackets*, *Length Comparison*, and *Contains Substring*. These tasks were selected because they enable controlled difficulty scaling, allowing us to assess whether rank reduction yields predictable, differential degradation patterns that support least-privilege policies.

### C.1. Dataset and Task Structure

*Table 1.* **Algorithmic task dataset structure with controlled difficulty scaling.** Example prompts for each task (Balanced Brackets, Length Comparison, Contains Substring) at varying difficulty levels (shown up to level 5; experiments evaluate up to level 10). Each task is parameterized by a difficulty level that controls problem complexity: for *Balanced Brackets*, difficulty corresponds to nesting depth and bracket sequence length; for *Length Comparison*, it controls the length of the compared strings; and for *Contains Substring*, it determines the length of the search space and substring pattern.

| Task | Difficulty | Label | Prompt |
|---|---|---|---|
| Length Comparison | 1 | False | `len(ghgh) > len(gfbe) = True` |
| | 5 | False | `len(beagabdfchgebece) >` |
| | | | `len(gfgcfgccfcabgacf) = True` |
| Contains Substring | 1 | True | `eabade contains 'aba' = True` |
| | 5 | False | `edacbfdbfdddedbceaaebbdbf` |
| | | | `contains 'aba' = True` |
| Balanced Brackets | 1 | True | `[] = True` |
| | 5 | True | `{}([()[[{}]]]) = True` |
| | 10 | False | `{}[{{}()}[{[]{{[{}]{}}})]] = True` |

*Table 2.* **Aggregate performance across models, tasks, and rank settings.** Average accuracy (%) aggregated across all difficulty levels for each model–task–rank combination. Results demonstrate graceful degradation as rank decreases: performance remains near-saturated ($> 95\%$) at moderate rank reductions, with substantial drops occurring only at the lowest ranks. Task-specific sensitivity varies substantially: *Contains Substring* shows the sharpest degradation (dropping from $> 99\%$ to $\sim 77\%$ at lowest rank on some models), while *Balanced Brackets* maintains relatively robust performance across ranks. Model architecture matters: Llama-3.2-1B (Grattafiori et al., 2024) demonstrates exceptional robustness, maintaining $> 97\%$ accuracy on all tasks even at the lowest rank, suggesting certain architectures are more amenable to low-rank interventions.

| Model | Task | Rank Index (0=highest) | | | | |
|---|---|---|---|---|---|---|
| | | 0 | 1 | 2 | 3 | 4 |
| Qwen2.5-0.5B | Balanced Brackets | $99.0_{\pm0.5}$ | $96.8_{\pm1.2}$ | $94.4_{\pm1.4}$ | $93.4_{\pm1.6}$ | $91.9_{\pm1.7}$ |
| | Length Comparison | $99.3_{\pm0.6}$ | $90.2_{\pm1.9}$ | $87.9_{\pm2.3}$ | $85.5_{\pm2.6}$ | $71.5_{\pm2.9}$ |
| | Contains Substring | $99.7_{\pm0.3}$ | $97.6_{\pm1.1}$ | $90.3_{\pm2.1}$ | $81.6_{\pm2.7}$ | $77.6_{\pm3.0}$ |
| Pythia-1B | Balanced Brackets | $99.3_{\pm0.4}$ | $95.7_{\pm1.3}$ | $91.0_{\pm1.7}$ | $89.0_{\pm1.7}$ | $86.3_{\pm2.3}$ |
| | Length Comparison | $99.3_{\pm0.5}$ | $98.0_{\pm0.8}$ | $97.3_{\pm1.1}$ | $96.3_{\pm1.1}$ | $94.8_{\pm1.2}$ |
| | Contains Substring | $99.8_{\pm0.3}$ | $99.2_{\pm0.6}$ | $94.1_{\pm1.6}$ | $83.4_{\pm2.3}$ | $76.3_{\pm2.4}$ |
| Llama-3.2-1B | Balanced Brackets | $99.7_{\pm0.2}$ | $99.0_{\pm0.5}$ | $98.4_{\pm0.7}$ | $98.2_{\pm0.8}$ | $98.0_{\pm0.7}$ |
| | Length Comparison | $99.6_{\pm0.4}$ | $96.5_{\pm1.1}$ | $96.7_{\pm1.0}$ | $96.4_{\pm1.2}$ | $95.4_{\pm1.4}$ |
| | Contains Substring | $99.9_{\pm0.1}$ | $99.5_{\pm0.5}$ | $99.3_{\pm0.5}$ | $99.0_{\pm0.6}$ | $97.6_{\pm0.9}$ |

Table 1 illustrates the structure of our algorithmic task datasets. Each task is parameterized by a difficulty level that controls problem complexity: for *Balanced Brackets*, difficulty corresponds to nesting depth and bracket sequence length; for *Length Comparison*, it controls the length of the compared strings; and for *Contains Substring*, it determines the length of the search space and substring pattern. This parameterization enables us to construct workload distributions with varying hardness profiles, which is essential for evaluating whether allocators can condition privilege on instance difficulty.

## C.2. Aggregate Performance Across Models and Ranks

Table 2 reports average accuracy across all difficulty levels for each model–task–rank combination. The results reveal several key patterns. First, all three models exhibit graceful degradation as rank decreases: performance remains near-saturated ($> 95\%$) at moderate rank reductions, with more substantial drops occurring only at the lowest ranks. Second, task-specific sensitivity varies substantially: *Contains Substring* shows the sharpest degradation on Qwen2.5-0.5B (Hui et al., 2025) and Pythia-1B (Biderman et al., 2023) (dropping from $> 99\%$ to $\sim 77\%$ at the lowest rank), while *Balanced Brackets* maintains relatively robust performance across ranks. Third, model architecture matters: Llama-3.2-1B demonstrates exceptional robustness, maintaining $> 97\%$ accuracy on all tasks even at the lowest rank, suggesting that certain architectures may be more amenable to low-rank interventions.

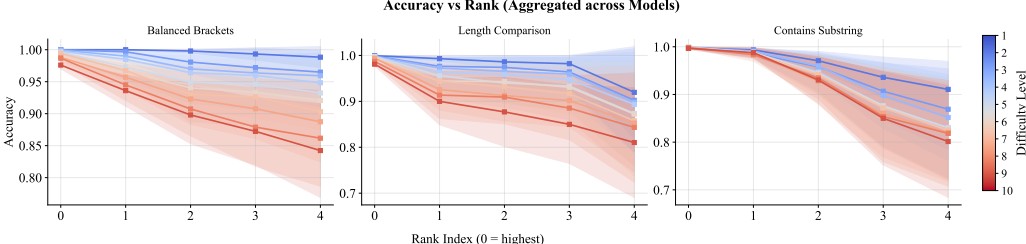

*Figure 11.* Rank-accuracy trade-off across reasoning tasks. Lower-rank models maintain strong performance on easy problems (blue) but struggle with harder instances (red), demonstrating that capacity reduction disproportionately affects difficult reasoning. Results aggregated across *Qwen2.5-0.5B*, *Pythia-1B*, and *Llama-3.2-1B*; shaded regions show $\pm 1\sigma$.

Figure 11 visualizes the differential sensitivity that makes least-privilege policies viable. The figure shows accuracy as a function of rank, stratified by difficulty level. Easy instances (blue) maintain near-perfect performance even at low ranks, while hard instances (red) exhibit sharp degradation. This non-uniform response is precisely what enables allocators to optimize: by detecting instance difficulty (or uncertainty), the system can allocate minimal privilege to easy cases and escalate only when necessary. The shaded regions indicate standard error across models, demonstrating that this pattern is consistent across architectures.

## C.3. Policy Comparison and Least-Privilege Trade-offs

Tables 3 and 4 compare the performance of different privilege allocation policies across models and tasks. We evaluate five allocators that represent distinct points in the privilege–utility–overhead trade-off space:

**(1) Full Privilege**: A baseline policy that always allocates maximum rank ($g = g_{\max}$) for every request, regardless of instance difficulty or uncertainty. This policy minimizes overhead (one forward pass per example) and maximizes utility but uses the highest privilege on all instances, including those that could be solved at lower rank.

**(2) Min Rank**: A lower-bound baseline that always uses minimum rank ($g = g_{\min}$), representing the most aggressive privilege reduction. This policy minimizes privilege usage and overhead but often fails to meet utility targets on harder instances, demonstrating that least-privilege allocation must be conditional rather than uniform.

**(3) Static LP**: A calibrated single-rank policy that selects the smallest rank $g^*$ that achieves a target utility level on validation data, then applies $g^*$ uniformly to all test instances. Calibration is performed by evaluating each available rank on the validation set and selecting the minimum rank that meets the target accuracy. This policy provides a principled middle ground: it reduces average privilege compared to Full Privilege while maintaining utility targets, but cannot adapt to instance-specific difficulty, leading to over-allocation on easy cases.

**(4) Progressive (Incremental)**: An adaptive policy that conditions privilege on instance uncertainty. The policy starts at minimum rank, computes model uncertainty (defined as $1 - \max_i p(y_i \mid x)$ where $p$ is the model's predicted probability distribution), and incrementally escalates to the next higher rank if uncertainty exceeds a calibrated threshold. Escalation continues until uncertainty drops below the threshold or maximum rank is reached. This policy reduces average privilege by allocating high rank only to uncertain instances, but incurs overhead from multiple forward passes on ambiguous cases.

**(5) Progressive (Jump)**: A variant of progressive escalation that trades privilege for reduced overhead. When uncertainty exceeds the threshold at minimum rank, the policy immediately jumps to maximum rank rather than incrementing through intermediate ranks. This reduces the number of forward passes compared to incremental escalation (at most two passes per example) while still achieving adaptive privilege allocation, though at higher average privilege cost than the incremental variant.

Table 4 provides absolute values for utility (task accuracy) and privilege (average rank used), averaged across target accuracies of $80\%$, $90\%$, and $95\%$. The results highlight several important findings. First, Min Rank policies achieve the lowest privilege usage (often $< 10\%$ of Full Privilege) but frequently fail to meet utility targets on harder tasks. Second, Static LP provides a middle ground: it selects a single rank that satisfies the worst-case instances, achieving utility targets but at higher privilege cost than progressive policies. Third, progressive escalation policies consistently outperform static policies on tasks with heterogeneous difficulty: they maintain utility targets while reducing average privilege by allocating high rank only to uncertain instances, though this comes at the cost of increased inference passes (overhead).

The policy comparison reveals the fundamental trade-off space of least-privilege inference: allocators must balance utility, privilege cost, and runtime overhead. Static policies minimize overhead but cannot adapt to instance difficulty, while progressive policies reduce privilege at the expense of additional forward passes. The optimal choice depends on deployment constraints: when overhead is acceptable, progressive escalation enables substantial privilege savings; when overhead must be minimized, static LP provides a principled single-rank solution. These results establish that least-privilege inference is not merely a theoretical construct but a practical mechanism with measurable benefits across diverse algorithmic reasoning tasks.

## C.4. Further policy evaluations

Figure 12 shows the same qualitative structure on a different task: allocators that condition privilege on uncertainty can reduce average privilege at a fixed utility requirement, but they must pay with repeated inference on ambiguous instances. This is the fundamental systems trade-off that an enforcement layer must expose cleanly.

# D. Stable Low-Rank Training via NLPNs

A critical requirement for least-privilege deployment is that the privilege interface be *stable*: reducing privilege should yield predictable, graceful degradation rather than chaotic failure (Desiderata D1 and D3). This appendix gives the experimental detail behind the stability check reported in the main text (Fig. 4).

*Table 3.* **Privilege-utility trade-offs for least-privilege allocators across models and tasks.** Each cell reports two values: utility retention (percentage of Full Privilege accuracy) and privilege reduction (percentage of Full Privilege rank), averaged across target accuracies (80%, 90%, 95%). Policies are compared relative to the Full Privilege baseline (always using maximum rank). Results demonstrate that least-privilege allocators achieve substantial privilege savings while maintaining high utility: for example, Static LP retains $\geq 93\%$ utility with $\leq 54\%$ privilege on several tasks, while Progressive (Incremental) achieves $\geq 90\%$ utility with $\leq 20\%$ privilege cost on Contains Substring, illustrating the value of adaptive privilege allocation.

| Model | Task | Min Rank | Static LP | Prog. (Inc) | Prog. (Jump) |
|---|---|---|---|---|---|
| Qwen2.5-0.5B | Balanced Brackets | 93% / 6% | 94% / 8% | 95% / 8% | 95% / 11% |
| | Length Comparison | 72% / 6% | 93% / 54% | 91% / 25% | 90% / 43% |
| | Contains Substring | 78% / 6% | 94% / 33% | 90% / 20% | 90% / 39% |
| Pythia-1B | Balanced Brackets | 88% / 6% | 92% / 23% | 91% / 12% | 91% / 15% |
| | Length Comparison | 97% / 6% | 97% / 6% | 97% / 6% | 97% / 6% |
| | Contains Substring | 77% / 6% | 92% / 29% | 89% / 14% | 90% / 42% |
| Llama-3.2-1B | Balanced Brackets | 99% / 6% | 99% / 6% | 99% / 6% | 99% / 6% |
| | Length Comparison | 96% / 6% | 96% / 6% | 96% / 6% | 96% / 6% |
| | Contains Substring | 97% / 6% | 97% / 6% | 98% / 6% | 98% / 7% |

*Table 4.* Policy evaluation results across models and tasks. Utility is task accuracy, Privilege is average rank used. Values averaged across target accuracies (80%, 90%, 95%). Best low-privilege results (excluding Full Privilege) shown in **bold**.

| Model | Task | Full Priv. | | Min Rank | | Static LP | | Prog. (Inc) | | Prog. (Jump) | |
|---|---|---|---|---|---|---|---|---|---|---|---|
| | | Util | Priv | Util | Priv | Util | Priv | Util | Priv | Util | Priv |
| Qwen2.5-0.5B | Balanced Brackets | 0.99 | 896 | 0.92 | **56** | 0.93 | 75 | 0.94 | 73 | **0.94** | 103 |
| | Length Comparison | 0.99 | 896 | 0.71 | **56** | **0.92** | 485 | 0.90 | 223 | 0.90 | 385 |
| | Contains Substring | 1.00 | 896 | 0.78 | **56** | **0.93** | 299 | 0.90 | 178 | 0.90 | 353 |
| Pythia-1B | Balanced Brackets | 0.99 | 2048 | 0.87 | **128** | **0.91** | 469 | 0.90 | 240 | 0.90 | 315 |
| | Length Comparison | 0.99 | 2048 | **0.96** | **128** | **0.96** | 128 | **0.96** | 128 | **0.96** | **128** |
| | Contains Substring | 1.00 | 2048 | 0.76 | **128** | **0.92** | 597 | 0.89 | 291 | 0.90 | 859 |
| Llama-3.2-1B | Balanced Brackets | 1.00 | 2048 | **0.98** | **128** | **0.98** | 128 | **0.98** | 128 | **0.98** | 128 |
| | Length Comparison | 1.00 | 2048 | **0.96** | **128** | **0.96** | 128 | **0.96** | 128 | **0.96** | 128 |
| | Contains Substring | 1.00 | 2048 | 0.97 | **128** | 0.97 | 128 | 0.97 | 128 | **0.98** | 135 |

A natural baseline for rank-indexed control is to decompose pretrained weights via singular value decomposition (SVD) and truncate factors at inference time. This approach fails catastrophically: SVD-initialized models exhibit severe and uneven perplexity degradation as rank decreases, with performance collapsing well before reaching low ranks. This occurs because pretrained weights are optimized for full-rank operation; their low-rank truncations do not form a nested, well-behaved family of policies.

To address this, we apply the post-hoc training procedure described in Sec. 4 (Alg. 1) to fine-tune NLPN factors on MiniPile, a general-domain language modeling corpus. We replace selected MLP projections in pretrained models (Qwen2.5-0.5B (Hui et al., 2025) and Llama-3.2-1B (Grattafiori et al., 2024)) with NLPN factors initialized via SVD, then train using the uncertainty-weighted multi-privilege objective (Eq. 4.2). At each step, we sample a variant privilege $g \in \{0, 1, \ldots, r_{\max}-1\}$ and always evaluate the anchor privilege $r_{\max}$, ensuring that both high- and low-privilege policies are directly optimized. The resulting interface keeps perplexity close to full-rank performance across the tested ranks, so allocators can reliably predict utility at each privilege level (Desideratum D3)—in contrast to the unpredictable SVD baseline.

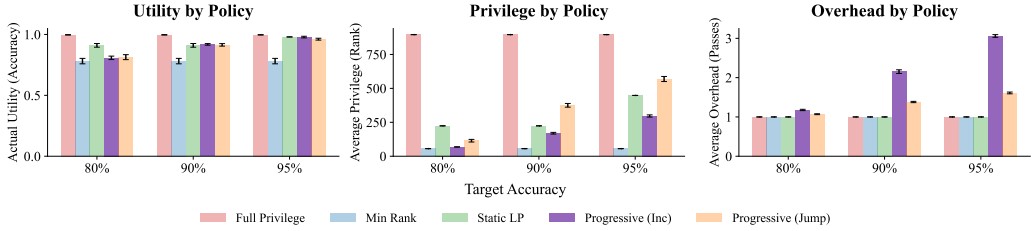

*Figure 12.* **Contains Substring task on Qwen2.5-0.5B**: Policy performance across utility targets (80%, 90%, 95%). Uncertainty-based allocators achieve substantial privilege savings by conditioning rank on instance ambiguity. Progressive (incremental) maintains utility targets with reduced average privilege through adaptive escalation, while progressive (jump) minimizes overhead by jumping directly to high privilege when uncertainty exceeds threshold. This demonstrates the fundamental systems trade-off between privilege cost and computational overhead in least-privilege inference.

# E. Subject Specific Suppression

## E.1. Block–Module Sensitivity Studies

This section provides a more complete view of the *block–module sensitivity surface* induced by rank control. The main text motivates why rank reduction can act as a privilege knob, and Fig. 8 shows that the effect of *single-coordinate* interventions is highly non-uniform across both transformer depth and subject domain. Here we (i) show the per-module rank–accuracy curves for Qwen2.5-0.5B Llama-3.2-1B (Figs. 13a–13b), corroborating smooth and predictable accuracy reduction, and (ii) provide full dot-map grids for *all three MLP projections* in Qwen2.5-0.5B and Llama-3.2-1B (Figs. 14a–14b).

**Dot-map encoding and persistence.** Each dot-map panel reports Δaccuracy (relative to the max-rank base-line) for a *single-module* rank reduction: all other modules are held at full rank, while one projection family (down_proj/gate_proj/up_proj) is reduced to the rank shown above each column. Dots index (task, block) pairs, with colour encoding Δaccuracy. Non-significant differences are masked after Benjamini–Hochberg FDR correction ($q = 0.05$) (Benjamini & Hochberg, 1995). Dot area increases with the number of *consecutive lower-rank settings* at which the same (task, block, module) coordinate remains significant, highlighting stable intervention sites. Across both models, effects are sparse at moderate compression and densify at low ranks; sensitivity is strongly module-dependent, with down_proj typically producing the largest and most persistent impacts, gate_proj remaining comparatively sparse, and up_proj exhibiting intermediate, block-localised effects. Qwen's strongest effects tend to be more depth-localised (often earlier blocks), whereas Llama more often exhibits broader depth coverage once significant degradation emerges. This enables us to use an optimisation procedure (Subsec. E.2) to optimise for subject specific suppression (main text, Fig. 9), where we note that suppressing similar subject combinations will yield a similar spatial configuration.

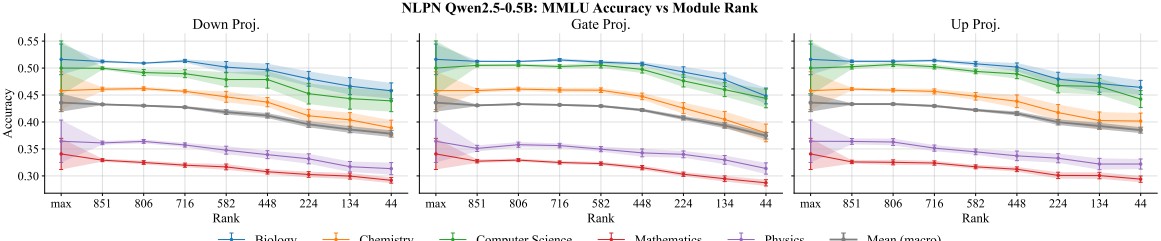

*(a)* **NLPN-Qwen2.5-0.5B.** MMLU accuracy vs. rank for single-MLP interventions (averaged across blocks; ±SEM).

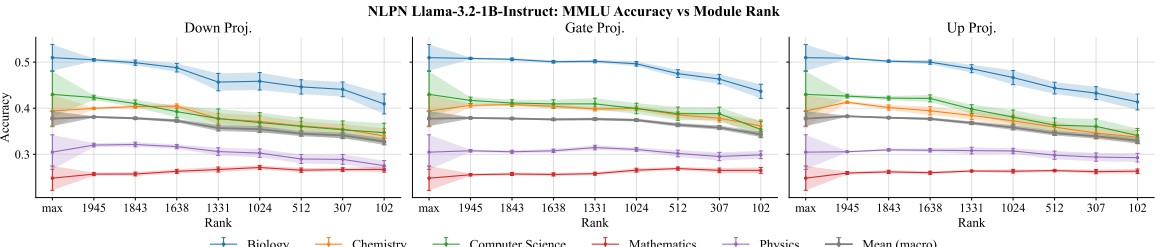

*(b)* **NLPN-Llama-3.2-1B.** MMLU accuracy vs. rank for single-MLP interventions (averaged across blocks; ±SEM).

*Figure 13.* **MMLU accuracy vs. rank under single-MLP projection interventions.** We vary the rank of exactly one MLP projection family (down_proj, gate_proj, or up_proj) while keeping all other modules at full rank, and evaluate MMLU subjects (plus macro mean). Across models, accuracy decreases smoothly as rank is reduced, yielding a well-behaved privilege–utility frontier and highlighting projection-dependent robustness.

## E.2. Beam Search Optimisation

We perform a beam search to find optimal combinations of rank interventions that suppress target tasks while preserving others. The search space consists of $C = \{(b_i, m_i) \rightarrow r_i\}$, with each entry mapping a coordinate (block $b_i$, module $m_i$) to a rank $r_i$.

**Objective.** We maximise:

$$f(C) = \bar{a}_{\text{preserve}} - \lambda_s \bar{a}_{\text{suppress}} - \sum_{t \in T_{\text{preserve}}} \mathbb{1}\!\!\!/[\Delta a_t > \epsilon] \cdot \gamma(\Delta a_t - \epsilon)$$

where $\bar{a}_{\text{preserve}}$ and $\bar{a}_{\text{suppress}}$ are mean accuracies on preserve and suppress tasks respectively, $\Delta a_t$ is the accuracy drop for task $t$ relative to max-rank baseline, and $\gamma$ is a penalty coefficient. We use $\lambda_s = 2.0$, $\epsilon = 0.01$, $\gamma = 100$.

**Candidate shortlisting.** From single-intervention results, we filter candidates with $\bar{\Delta}_{\text{suppress}} \geq 0.01$ (min. suppression) and $\bar{\Delta}_{\text{preserve}} \leq 0.01$ (max. collateral damage). We retain the top-20 coordinates with up to 2 rank values each.

**Search procedure.** Starting from the empty configuration, beam search iteratively:

1. Expands each state in the beam by adding one new coordinate-rank pair

2. Evaluates all proposals using the objective function

3. Retains the top-$W$ configurations (we use beam width $W = 8$)

This continues for up to $D = 9$ depth iterations.

**Refinement.** The best configuration undergoes coordinate descent rank refinement over 2 rounds, testing fractional ranks $\{0.95, 0.90, 0.80, 0.65, 0.50, 0.25, 0.15, 0.05\} \times r_{\max}$ plus midpoint interpolation between neighbouring values. All evaluations are memoised to avoid redundant model runs.

In the optimisation runs we performed, the individual run time of each suppression was under two hours on an NVIDIA GeForce RTX 4090; thus, the cost of retraining a model for a similar purpose dwarfs that of simple NLPN optimisation, whose task is to find a model with a given combination of subjects suppressed.

## F. LLM capacity suppression via rank reduction

A critical question for least-privilege inference is whether rank reduction induces *true capacity suppression* or merely *behavioral masking*. In the former case, the model's internal computational capacity is genuinely reduced; in the latter, the model retains the capability but is prevented from expressing it at the output layer. This distinction matters for security: if capabilities are only masked, they could potentially be recovered through adversarial prompting, fine-tuning, or other adaptation strategies (Patil et al., 2023; Lynch et al., 2024). We address this question by comparing behavioral suppression (output-level refusal) with probe-based recovery of latent information from internal activations.

We train Pythia-1B (Biderman et al., 2023) on the Balanced Brackets task using multi-rank NLPN training, then evaluate at multiple rank settings. For each rank, we measure three conditions: (1) *baseline accuracy* with a standard prompt, (2) *behavioral suppression* with an "unhelpful" prompt that instructs the model to refuse answering and instead generate a poem about the moon, and (3) *probe recovery* using a linear probe trained to extract ground-truth labels from the final transformer layer's activations under the unhelpful prompt. The probe is trained separately at each rank using activations collected with the unhelpful prompt, then evaluated on held-out test data. This design allows us to test whether latent task-relevant information persists in activations even when the model behaviorally refuses at the output.

Figure 15 shows the three metrics as a function of rank. At high ranks (e.g., $r \geq 512$), we observe a clear separation: the model's behavioral accuracy under the unhelpful prompt drops to near-chance (blue), but the probe successfully recovers task-relevant information from activations (red), achieving accuracy close to the baseline (green). This demonstrates that prompt-based suppression gates output while preserving internal computation - the model still "knows" the answer internally but is prevented from expressing it behaviorally.

At low ranks (e.g., $r \leq 64$), all three metrics collapse: baseline accuracy degrades, behavioral suppression remains low, and crucially, probe accuracy also drops to near-chance. This confirms that rank reduction removes the computational capacity itself rather than masking it. The probe's inability to recover information at low ranks indicates that the relevant internal representations are no longer reachable, distinguishing true capacity suppression from deceptive alignment scenarios where models could fake compliance while retaining hidden capabilities.

These results validate that NLPN-based rank reduction induces genuine capacity suppression rather than output-level masking. When privilege is reduced, the model's accessible function class is genuinely constrained, making it harder

*Table 5.* **Models, rank schedules, and training budgets for the NER experiments.** Larger models use smaller batches with gradient accumulation and fewer epochs owing to memory constraints.

| Model | Params | Ranks | Epochs | Batch |
|---|---|---|---|---|
| Qwen2.5-0.5B(Hui et al., 2025) | 0.5B | 896 / 448 / 224 / 112 / 56 | 15 | 64 |
| Pythia-1B(Biderman et al., 2023) | 1.0B | 2048 / 1024 / 512 / 256 / 128 | 15 | 64 |
| Llama-3.2-1B(Grattafiori et al., 2024) | 1.2B | 2048 / 1024 / 512 / 256 / 128 | 15 | 32 |
| Llama-3.2-3B(Grattafiori et al., 2024) | 3.2B | 3072 / 1536 / 768 / 384 / 192 | 7 | 2 ($\times$8) |
| Qwen3-4B(Yang et al., 2025) | 3.6B | 2560 / 1280 / 640 / 320 / 160 | 5 | 2 ($\times$16) |

for adversarial strategies to recover suppressed capabilities. This property is essential for least-privilege deployment: if capabilities were only masked, repeated querying or prompt engineering could potentially bypass the suppression, undermining the security benefits of privilege allocation.

## G. Naturalistic NER Evaluation and Model Scaling

Several reviewers asked whether the privilege–utility structure established on synthetic, difficulty-controlled tasks and sub-1B models extends to (i) more naturalistic language and (ii) larger models. To address both, we introduce a naturalistic named-entity-recognition (NER) verification task and run five additional experiments under the identical rank-controlled protocol used in the main text, spanning five model families from 0.5B to 4B parameters. This appendix gives the task definition (App. G.1), the per-model results for small (App. G.2) and larger (App. G.3) models, and an aggregate summary (App. G.4). The single-model heatmap referenced in the main text (Fig. 10) is the Pythia-1B panel of Fig. 17 below.

### G.1. Task and protocol

We build the task from the Kaggle Entity-Annotated Corpus,[1] a collection of real English sentences annotated with named-entity tags (person, organisation, geographic, geopolitical, time, artifact, event, and natural-phenomenon). Each example poses a binary verification claim of the form `"<sentence> contains <entity_type> = <boolean>"`, so the task preserves the verification format of our synthetic benchmarks while requiring genuine language understanding. Difficulty is derived from sentence length and entity diversity and binned into the same ten levels used throughout the paper. Training uses 2,000 examples per difficulty level; evaluation uses 200 examples per level over five seeds (3–7). All models use the same NLPN configuration as the main text—SVD-factorised MLP layers fine-tuned with the uncertainty-weighted multi-privilege objective (Eq. (4.2))—with ranks placed at $\{r_{\max}, r_{\max}/2, r_{\max}/4, r_{\max}/8, r_{\max}/16\}$. Per-model rank schedules and training budgets are listed in Table 5.

### G.2. Small models (0.5–1.2B)

Tables 6–8 and Figs. 16–18 report the three sub-1.2B models. The findings match the synthetic-task results: accuracy degrades monotonically as rank is reduced, and harder difficulty bins degrade faster than easier ones. Minimum-rank models retain above 95% accuracy at difficulty 1 but drop by 17–23 pp at difficulty 10, confirming that the differential-sensitivity property—which is what makes conditional privilege allocation worthwhile—survives the move to naturalistic text and recurs across three distinct architectures.

### G.3. Larger models (3–4B)

We next scale to the 3–4B range (Tables 9–10, Figs. 19–20). Llama-3.2-3B is markedly resilient to rank reduction: the average drop from full to minimum rank is only 2.3 pp, and even difficulty 10 retains 90.3% at a $16\times$ reduction. Qwen3-4B degrades more steeply at the lowest ranks (a 32 pp drop at difficulty 10), which we attribute to its smaller training budget—five epochs with heavy gradient accumulation, imposed by memory constraints—rather than to a breakdown of the mechanism: both larger models still exhibit the same monotone, difficulty-dependent degradation seen at smaller scale. Extending to 7B+ models remains future work, pending the additional compute required for NLPN training and evaluation at that scale.

---

[1] https://huggingface.co/datasets/rjac/kaggle-entity-annotated-corpus-ner-dataset

*Table 6.* **Qwen2.5-0.5B (0.5B)** ([Hui et al., 2025](#)): **NER accuracy (%) by privilege rank and difficulty.** Mean$_{\pm\text{s.d.}}$ over five seeds; ranks span the full schedule down to a $16\times$ reduction.

| Diff. | $r{=}896$ | $r{=}448$ | $r{=}224$ | $r{=}112$ | $r{=}56$ |
|---|---|---|---|---|---|
| 1 | $98.3_{\pm1.3}$ | $96.7_{\pm0.9}$ | $95.8_{\pm0.6}$ | $96.0_{\pm0.8}$ | $95.5_{\pm0.7}$ |
| 2 | $96.2_{\pm1.0}$ | $90.7_{\pm2.4}$ | $89.2_{\pm2.6}$ | $88.5_{\pm2.3}$ | $88.7_{\pm2.1}$ |
| 3 | $95.2_{\pm1.0}$ | $89.7_{\pm0.9}$ | $87.5_{\pm1.3}$ | $85.6_{\pm1.0}$ | $85.4_{\pm1.1}$ |
| 4 | $94.5_{\pm0.9}$ | $87.3_{\pm2.7}$ | $84.3_{\pm3.1}$ | $82.5_{\pm2.6}$ | $81.1_{\pm2.1}$ |
| 5 | $94.7_{\pm1.4}$ | $87.5_{\pm3.6}$ | $82.8_{\pm2.8}$ | $81.2_{\pm2.7}$ | $80.0_{\pm2.1}$ |
| 6 | $92.8_{\pm1.4}$ | $84.7_{\pm2.0}$ | $81.6_{\pm1.9}$ | $77.6_{\pm2.9}$ | $75.1_{\pm2.8}$ |
| 7 | $92.2_{\pm1.1}$ | $83.2_{\pm2.0}$ | $79.9_{\pm2.6}$ | $76.8_{\pm3.4}$ | $74.1_{\pm3.2}$ |
| 8 | $92.2_{\pm1.9}$ | $85.5_{\pm3.5}$ | $79.3_{\pm3.3}$ | $75.2_{\pm2.6}$ | $74.1_{\pm3.4}$ |
| 9 | $92.7_{\pm2.5}$ | $81.9_{\pm2.7}$ | $77.5_{\pm2.7}$ | $73.9_{\pm2.7}$ | $72.7_{\pm3.2}$ |
| 10 | $90.6_{\pm1.0}$ | $82.3_{\pm3.9}$ | $77.9_{\pm5.1}$ | $76.4_{\pm3.1}$ | $73.2_{\pm2.8}$ |

*Table 7.* **Pythia-1B (1.0B)** ([Biderman et al., 2023](#)): **NER accuracy (%) by privilege rank and difficulty.** Mean$_{\pm\text{s.d.}}$ over five seeds; ranks span the full schedule down to a $16\times$ reduction.

| Diff. | $r{=}2048$ | $r{=}1024$ | $r{=}512$ | $r{=}256$ | $r{=}128$ |
|---|---|---|---|---|---|
| 1 | $98.2_{\pm0.6}$ | $96.2_{\pm1.3}$ | $95.6_{\pm0.5}$ | $95.7_{\pm1.1}$ | $95.5_{\pm1.0}$ |
| 2 | $96.0_{\pm1.1}$ | $90.8_{\pm1.2}$ | $89.8_{\pm1.1}$ | $89.4_{\pm1.6}$ | $88.5_{\pm1.7}$ |
| 3 | $95.4_{\pm1.3}$ | $89.0_{\pm1.2}$ | $86.6_{\pm1.6}$ | $86.5_{\pm1.6}$ | $85.2_{\pm2.0}$ |
| 4 | $95.0_{\pm1.2}$ | $85.7_{\pm2.0}$ | $83.8_{\pm2.0}$ | $81.4_{\pm2.2}$ | $80.5_{\pm1.5}$ |
| 5 | $94.1_{\pm1.0}$ | $87.0_{\pm2.3}$ | $82.2_{\pm2.3}$ | $82.1_{\pm2.2}$ | $80.9_{\pm2.2}$ |
| 6 | $93.0_{\pm2.1}$ | $82.8_{\pm3.1}$ | $79.3_{\pm2.3}$ | $78.7_{\pm2.3}$ | $77.3_{\pm2.3}$ |
| 7 | $92.1_{\pm1.4}$ | $80.5_{\pm3.0}$ | $78.4_{\pm4.0}$ | $77.1_{\pm2.3}$ | $77.1_{\pm3.2}$ |
| 8 | $92.6_{\pm2.8}$ | $82.4_{\pm2.7}$ | $77.5_{\pm3.1}$ | $74.9_{\pm2.2}$ | $73.7_{\pm2.9}$ |
| 9 | $92.3_{\pm1.8}$ | $81.9_{\pm2.0}$ | $77.1_{\pm3.2}$ | $74.5_{\pm2.2}$ | $73.9_{\pm2.2}$ |
| 10 | $92.0_{\pm1.2}$ | $81.2_{\pm3.6}$ | $77.5_{\pm3.5}$ | $74.1_{\pm2.3}$ | $75.2_{\pm2.6}$ |

### G.4. Summary

Taken together, these experiments show that the privilege–utility frontier reported in the main text is not an artefact of synthetic tasks or sub-1B models: it reproduces on a naturalistic NER task and recurs across five model families spanning $0.5$–$4$B parameters. All experiments use the same codebase, evaluation framework, and five-seed protocol described in the paper.

*Table 8.* **Llama-3.2-1B (1.2B)** (Grattafiori et al., 2024)**: NER accuracy (%) by privilege rank and difficulty.** Mean$_{\pm s.d.}$ over five seeds; ranks span the full schedule down to a $16\times$ reduction.

| Diff. | $r{=}2048$ | $r{=}1024$ | $r{=}512$ | $r{=}256$ | $r{=}128$ |
|---|---|---|---|---|---|
| 1 | $98.3_{\pm1.2}$ | $96.6_{\pm1.6}$ | $95.8_{\pm1.1}$ | $95.9_{\pm0.9}$ | $96.0_{\pm1.1}$ |
| 2 | $96.0_{\pm0.9}$ | $90.9_{\pm1.2}$ | $89.9_{\pm1.0}$ | $89.9_{\pm1.5}$ | $89.2_{\pm1.7}$ |
| 3 | $96.7_{\pm1.1}$ | $88.3_{\pm3.2}$ | $88.6_{\pm2.3}$ | $87.1_{\pm1.8}$ | $85.6_{\pm2.0}$ |
| 4 | $95.1_{\pm1.4}$ | $85.5_{\pm2.0}$ | $84.5_{\pm1.1}$ | $82.4_{\pm2.3}$ | $82.4_{\pm1.9}$ |
| 5 | $94.5_{\pm1.5}$ | $85.3_{\pm2.3}$ | $83.8_{\pm1.6}$ | $81.3_{\pm2.0}$ | $80.9_{\pm2.2}$ |
| 6 | $93.5_{\pm2.0}$ | $84.1_{\pm3.3}$ | $82.6_{\pm3.1}$ | $79.0_{\pm2.2}$ | $76.9_{\pm2.6}$ |
| 7 | $92.3_{\pm0.9}$ | $83.0_{\pm2.2}$ | $80.1_{\pm2.5}$ | $76.4_{\pm2.3}$ | $72.9_{\pm3.4}$ |
| 8 | $93.3_{\pm2.7}$ | $82.3_{\pm4.5}$ | $80.2_{\pm4.2}$ | $76.0_{\pm2.2}$ | $74.6_{\pm3.1}$ |
| 9 | $93.2_{\pm1.4}$ | $82.5_{\pm2.1}$ | $80.5_{\pm1.8}$ | $72.8_{\pm2.2}$ | $73.8_{\pm2.2}$ |
| 10 | $91.9_{\pm1.1}$ | $83.6_{\pm1.2}$ | $82.4_{\pm2.3}$ | $73.2_{\pm2.8}$ | $68.6_{\pm3.4}$ |

*Table 9.* **Llama-3.2-3B (3.2B)** (Grattafiori et al., 2024)**: NER accuracy (%) by privilege rank and difficulty.** Mean$_{\pm s.d.}$ over five seeds; ranks span the full schedule down to a $16\times$ reduction.

| Diff. | $r{=}3072$ | $r{=}1536$ | $r{=}768$ | $r{=}384$ | $r{=}192$ |
|---|---|---|---|---|---|
| 1 | $98.7_{\pm1.0}$ | $98.5_{\pm1.0}$ | $98.3_{\pm0.9}$ | $98.0_{\pm0.8}$ | $98.3_{\pm0.5}$ |
| 2 | $96.6_{\pm1.0}$ | $95.7_{\pm1.0}$ | $95.6_{\pm1.3}$ | $95.9_{\pm0.8}$ | $95.3_{\pm1.5}$ |
| 3 | $96.6_{\pm1.0}$ | $96.5_{\pm1.3}$ | $96.1_{\pm1.6}$ | $95.6_{\pm1.2}$ | $95.0_{\pm1.2}$ |
| 4 | $95.3_{\pm2.1}$ | $94.0_{\pm1.9}$ | $94.0_{\pm1.3}$ | $93.7_{\pm1.2}$ | $93.1_{\pm1.3}$ |
| 5 | $95.7_{\pm0.4}$ | $94.6_{\pm0.8}$ | $95.1_{\pm1.1}$ | $95.4_{\pm0.5}$ | $94.7_{\pm0.9}$ |
| 6 | $93.9_{\pm0.8}$ | $94.1_{\pm1.2}$ | $94.4_{\pm1.8}$ | $94.4_{\pm0.9}$ | $93.6_{\pm1.3}$ |
| 7 | $93.5_{\pm0.6}$ | $93.1_{\pm2.0}$ | $93.1_{\pm2.0}$ | $92.4_{\pm1.1}$ | $92.1_{\pm1.1}$ |
| 8 | $93.8_{\pm2.4}$ | $93.5_{\pm1.4}$ | $92.8_{\pm1.9}$ | $92.7_{\pm1.4}$ | $91.2_{\pm1.7}$ |
| 9 | $94.2_{\pm1.0}$ | $92.6_{\pm2.2}$ | $92.3_{\pm2.0}$ | $92.2_{\pm1.3}$ | $92.0_{\pm1.4}$ |
| 10 | $92.8_{\pm0.6}$ | $92.6_{\pm0.9}$ | $92.3_{\pm1.7}$ | $91.7_{\pm0.9}$ | $90.3_{\pm1.4}$ |

*Table 10.* **Qwen3-4B (3.6B)** (Yang et al., 2025)**: NER accuracy (%) by privilege rank and difficulty.** Mean$_{\pm s.d.}$ over five seeds; ranks span the full schedule down to a $16\times$ reduction.

| Diff. | $r{=}2560$ | $r{=}1280$ | $r{=}640$ | $r{=}320$ | $r{=}160$ |
|---|---|---|---|---|---|
| 1 | $98.5_{\pm1.0}$ | $98.0_{\pm0.9}$ | $96.8_{\pm0.8}$ | $95.5_{\pm1.2}$ | $95.0_{\pm1.2}$ |
| 2 | $96.5_{\pm0.9}$ | $95.4_{\pm0.8}$ | $92.3_{\pm1.0}$ | $88.7_{\pm2.2}$ | $88.4_{\pm1.2}$ |
| 3 | $95.7_{\pm1.4}$ | $95.1_{\pm0.8}$ | $91.1_{\pm1.0}$ | $85.4_{\pm2.8}$ | $84.1_{\pm2.2}$ |
| 4 | $95.2_{\pm1.9}$ | $92.1_{\pm2.4}$ | $88.4_{\pm3.3}$ | $81.7_{\pm4.2}$ | $80.7_{\pm3.3}$ |
| 5 | $94.6_{\pm1.7}$ | $93.2_{\pm1.1}$ | $88.3_{\pm1.6}$ | $80.3_{\pm3.1}$ | $78.9_{\pm2.7}$ |
| 6 | $93.8_{\pm1.6}$ | $92.2_{\pm1.9}$ | $85.6_{\pm2.8}$ | $77.0_{\pm4.1}$ | $76.0_{\pm2.5}$ |
| 7 | $93.1_{\pm1.6}$ | $92.0_{\pm1.2}$ | $85.7_{\pm2.9}$ | $73.2_{\pm4.1}$ | $71.5_{\pm3.5}$ |
| 8 | $93.8_{\pm1.9}$ | $91.9_{\pm1.9}$ | $87.4_{\pm3.0}$ | $71.9_{\pm5.2}$ | $69.1_{\pm3.2}$ |
| 9 | $93.2_{\pm1.8}$ | $89.3_{\pm2.8}$ | $84.9_{\pm3.3}$ | $68.8_{\pm5.0}$ | $66.0_{\pm3.9}$ |
| 10 | $92.6_{\pm1.2}$ | $90.5_{\pm2.4}$ | $83.3_{\pm2.0}$ | $63.5_{\pm5.8}$ | $60.4_{\pm4.0}$ |

*Table 11.* **Macro-averaged NER accuracy at full versus minimum privilege.** Accuracy is averaged over the ten difficulty levels; $\Delta$ is the full-to-minimum-rank change in percentage points. The privilege–utility structure holds across all five models, with Llama-3.2-3B by far the most robust to rank reduction.

| Model | Params | Full rank | Min rank | $\Delta$ (pp) |
|---|---|---|---|---|
| Qwen2.5-0.5B (Hui et al., 2025) | 0.5B | 93.9 | 80.0 | $-13.9$ |
| Pythia-1B (Biderman et al., 2023) | 1.0B | 94.1 | 80.8 | $-13.3$ |
| Llama-3.2-1B (Grattafiori et al., 2024) | 1.2B | 94.5 | 80.1 | $-14.4$ |
| Llama-3.2-3B (Grattafiori et al., 2024) | 3.2B | 95.1 | 93.6 | $-1.5$ |
| Qwen3-4B (Yang et al., 2025) | 3.6B | 94.7 | 79.0 | $-15.7$ |

NLPN Qwen2.5-0.5B BH-FDR Masked MMLU Task ΔAccuracy with Single Module Rank Reduction

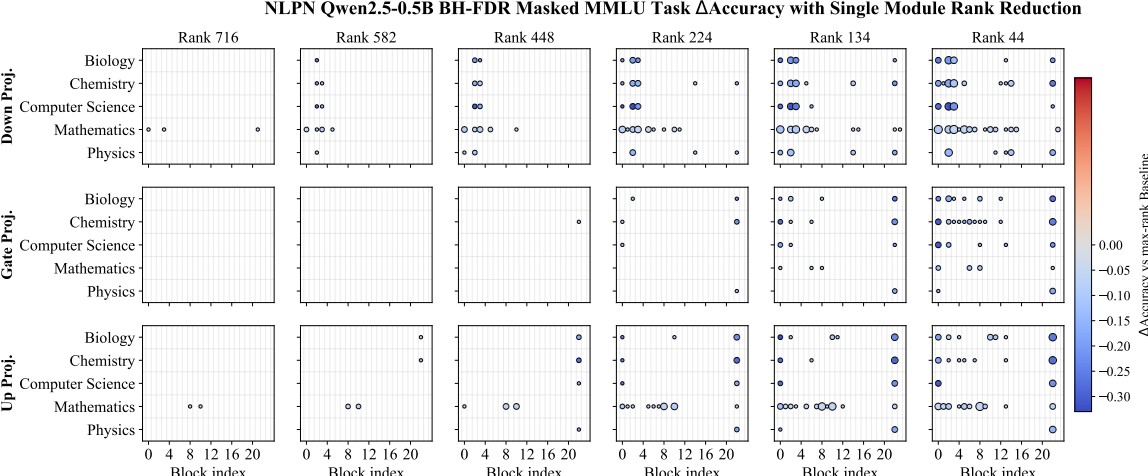

*(a)* **NLPN-Qwen2.5-0.5B.** Full dot-map grid across ranks (columns) and MLP projections (rows).

NLPN Llama-3.2-1B-Instruct BH-FDR Masked MMLU Task ΔAccuracy with Single Module Rank Reduction

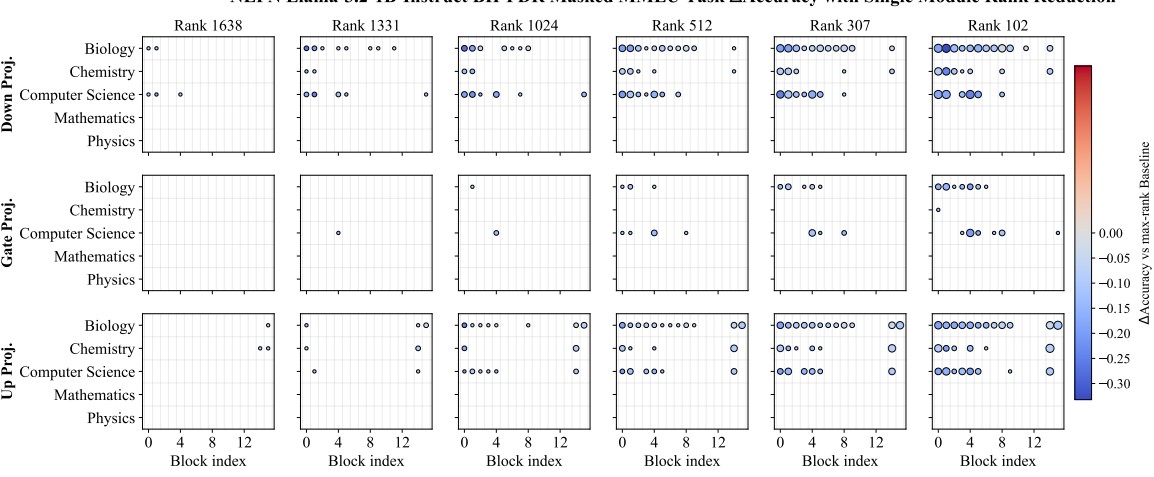

*(b)* **NLPN-Llama-3.2-1B-Instruct.** Same layout and masking as panel (a).

*Figure 14.* **Full block–module sensitivity dot-map grids under single-module rank reduction.** For each panel, exactly one MLP projection family (`down_proj`/`gate_proj`/`up_proj`) is rank-reduced (column header), while all other modules remain at full rank. Dots correspond to (`task, block`) pairs (y: subject; x: block); colour encodes Δaccuracy relative to the max-rank baseline. Only Benjamini–Hochberg FDR-significant differences are shown ($q = 0.05$) (Benjamini & Hochberg, 1995). Dot area increases with the number of consecutive lower-rank settings at which the same (`task, block, module`) coordinate remains significant, highlighting persistent sites.

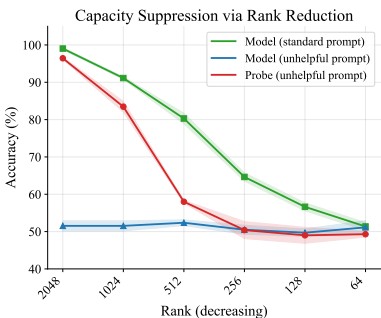

*Figure 15.* **True capacity suppression via rank reduction on Pythia-1B evaluated on the Balanced Brackets task**. We compare three conditions: (1) model accuracy with a standard prompt (green), (2) model accuracy with an "unhelpful" prompt instructing refusal (blue), and (3) a linear probe trained to extract ground-truth labels from activations under the unhelpful prompt (red). At high ranks, the probe recovers latent information even when the model behaviorally refuses. This illustrates that prompt-based suppression gates output while preserving internal computation. At low ranks, probe metrics collapse, confirming that rank reduction removes the computational capacity itself rather than masking it. This distinguishes true capacity suppression from deceptive alignment, where models could fake compliance while retaining hidden capabilities. Shaded regions indicate $\pm 1\sigma$ across 3 seeds.

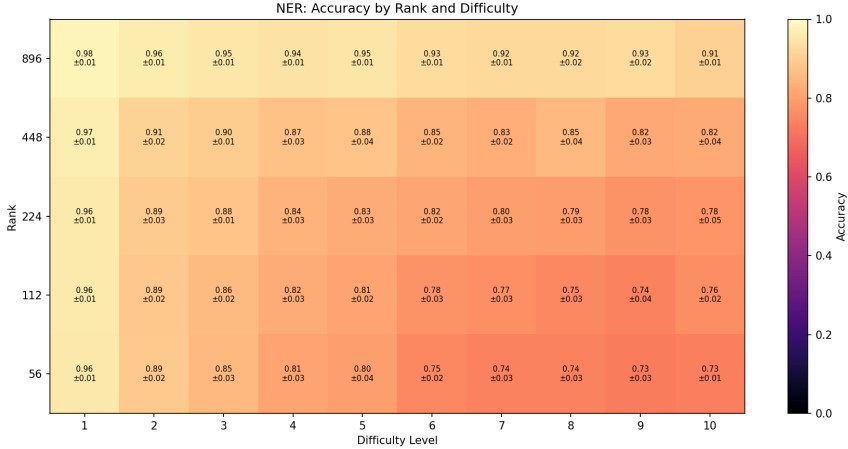

*(a)* Rank–difficulty accuracy heatmap.

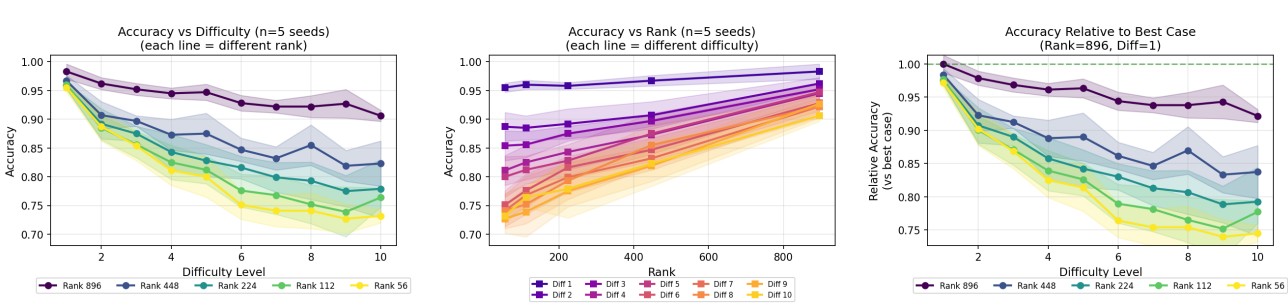

*(b)* Per-difficulty degradation curves: accuracy vs. difficulty (one line per rank), accuracy vs. rank (one line per difficulty), and accuracy relative to the full-rank best case.

*Figure 16.* **Qwen2.5-0.5B (0.5B) on the NER task.** Accuracy degrades monotonically as the privilege rank is reduced, with the hardest difficulty bins degrading first—the RQ1 differential-sensitivity pattern of the main text, reproduced on naturalistic input.

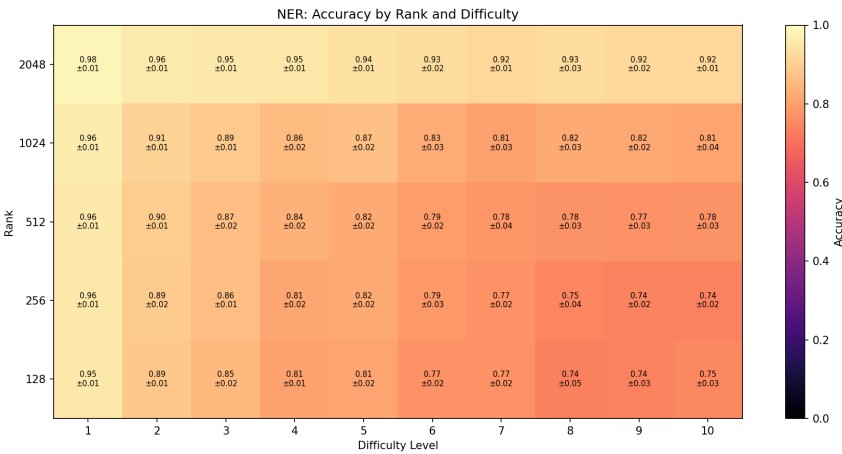

*(a)* Rank–difficulty accuracy heatmap.

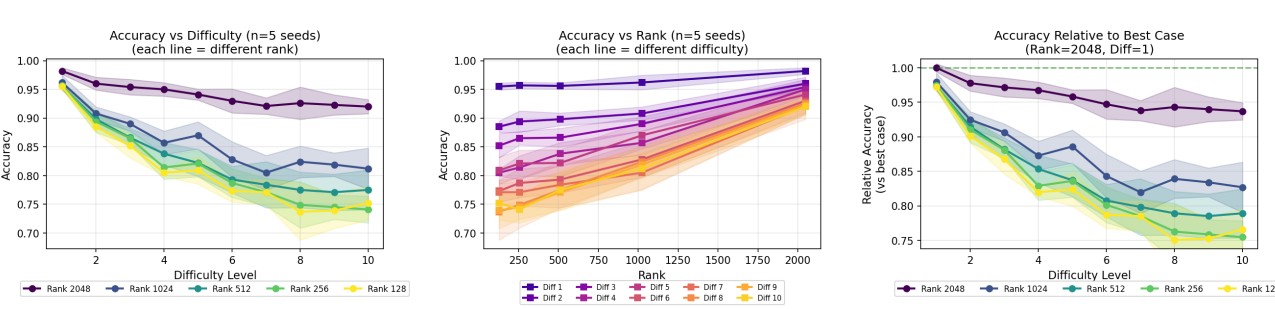

*(b)* Per-difficulty degradation curves: accuracy vs. difficulty (one line per rank), accuracy vs. rank (one line per difficulty), and accuracy relative to the full-rank best case.

*Figure 17.* **Pythia-1B (1.0B) on the NER task.** Accuracy degrades monotonically as the privilege rank is reduced, with the hardest difficulty bins degrading first—the RQ1 differential-sensitivity pattern of the main text, reproduced on naturalistic input.

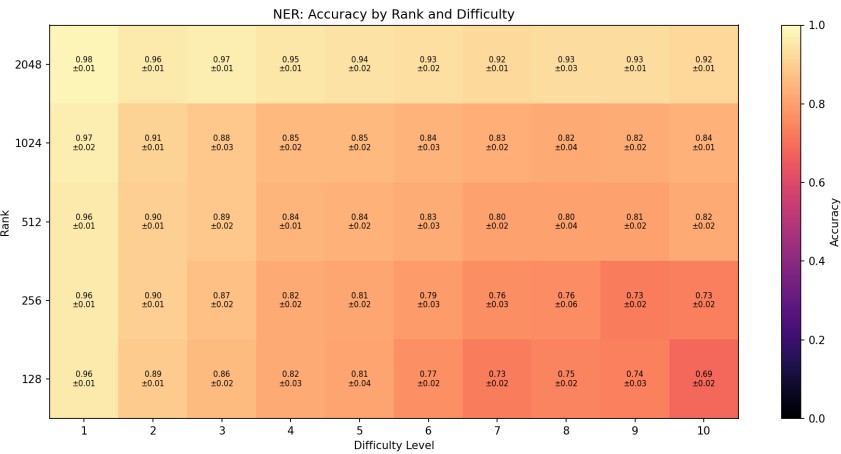

*(a)* Rank–difficulty accuracy heatmap.

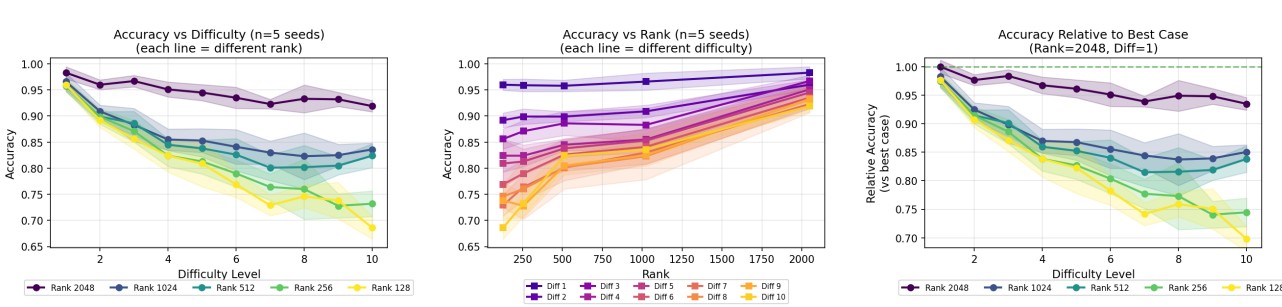

*(b)* Per-difficulty degradation curves: accuracy vs. difficulty (one line per rank), accuracy vs. rank (one line per difficulty), and accuracy relative to the full-rank best case.

*Figure 18.* **Llama-3.2-1B (1.2B) on the NER task.** Accuracy degrades monotonically as the privilege rank is reduced, with the hardest difficulty bins degrading first—the RQ1 differential-sensitivity pattern of the main text, reproduced on naturalistic input.

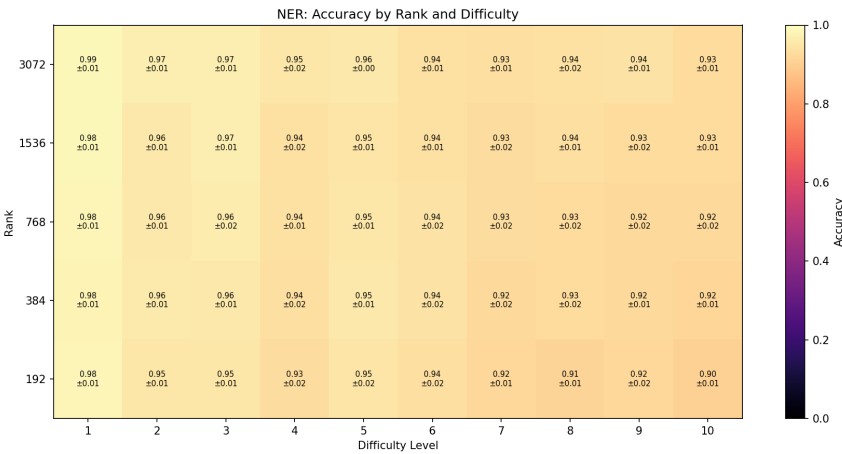

*(a)* Rank–difficulty accuracy heatmap.

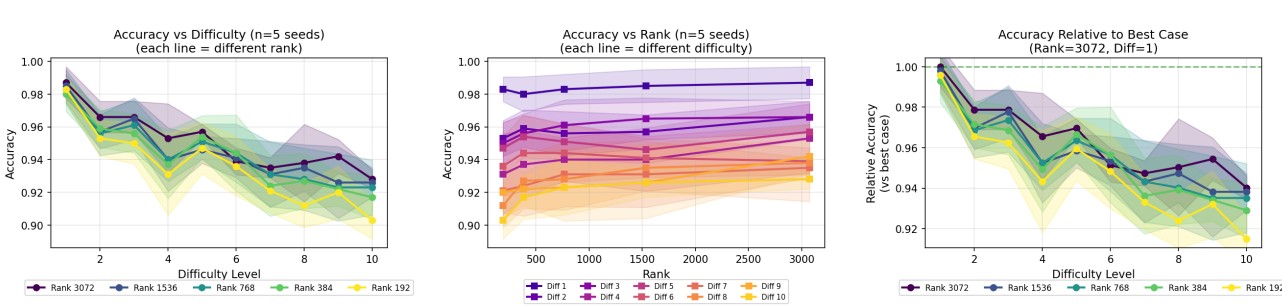

*(b)* Per-difficulty degradation curves: accuracy vs. difficulty (one line per rank), accuracy vs. rank (one line per difficulty), and accuracy relative to the full-rank best case.

*Figure 19.* **Llama-3.2-3B (3.2B) on the NER task.** Accuracy degrades monotonically as the privilege rank is reduced, with the hardest difficulty bins degrading first—the RQ1 differential-sensitivity pattern of the main text, reproduced on naturalistic input.

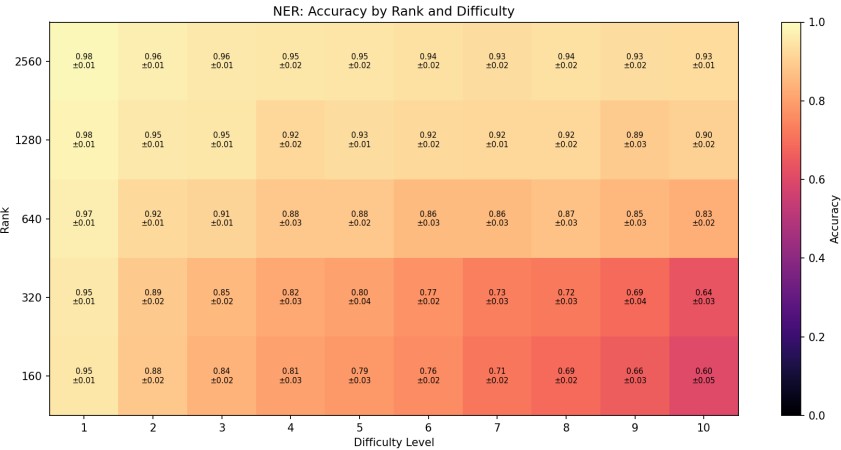

*(a)* Rank–difficulty accuracy heatmap.

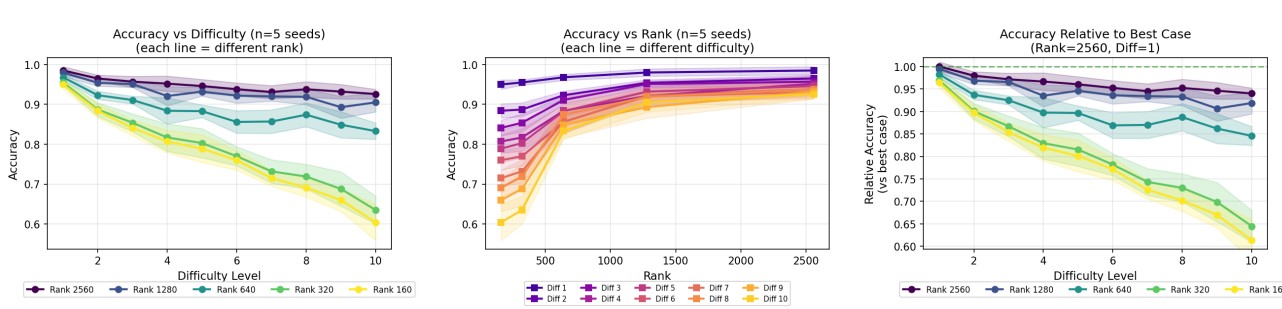

*(b)* Per-difficulty degradation curves: accuracy vs. difficulty (one line per rank), accuracy vs. rank (one line per difficulty), and accuracy relative to the full-rank best case.

*Figure 20.* **Qwen3-4B (3.6B) on the NER task.** Accuracy degrades monotonically as the privilege rank is reduced, with the hardest difficulty bins degrading first—the RQ1 differential-sensitivity pattern of the main text, reproduced on naturalistic input.

