# No More, No Less: Least-Privilege Language Models

## Abstract

Least privilege is a core security principle: grant each request only the minimum access needed to achieve its goal. Deployed language models almost never follow it, instead being exposed through a single API endpoint that serves all users and requests. This gap exists not because least privilege would be unhelpful—deployments would benefit greatly from reducing unnecessary capability exposure. The real obstacle is definitional and mechanistic: what does "access" mean inside a language model, and how can we enforce it without retraining or deploying multiple models? We take inspiration from least privilege in computer systems and define a class of models called *least-privilege language models*, where privilege is *reachable internal computation* during the forward pass. In this view, lowering privilege literally shrinks the model's accessible function class (as opposed to denying access via learned policies). We formalize deployment-time control as a monitor–allocator–enforcer stack, separating (i) request-time signals, (ii) a decision rule that allocates privilege, and (iii) an inference-time mechanism that selects privilege. We then propose *Nested Least-Privilege Networks*, a shape-preserving, rank-indexed intervention that provides a smooth, reversible control knob. We show that this knob yields policy-usable privilege–utility frontiers and enables selective suppression of targeted capabilities with limited collateral degradation across various policies. Most importantly, we see this as a defense of a completely new deployment paradigm which challenges the premise that we can only have output-level control of language models.

[1]Anonymous Institution, Anonymous City, Anonymous Region, Anonymous Country. Correspondence to: Anonymous Author <anon.email@domain.com>.

Preliminary work. Under review by the International Conference on Machine Learning (ICML). Do not distribute.

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

## 5. Illustrative studies

We now treat the enforcement surface as a control object and ask: *is it usable for least-privilege inference?* We frame our investigation as key research questions (**RQs**)[1].

---

[1]Our implementation, experimental code, and detailed results are publicly (and anonymously) available at https://anonymous.4open.science/r/llc-AC18

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

> 💡 **Insight 4**. There are block-level localizations that, when suppressed, act as privilege-reducing mechanisms on a subset of tasks while maintaining utility (performance) on other tasks.

## 5.5. RQ5: Can we directly *optimize* a model to reduce privilege on custom tasks?

We now care about directly optimizing a model such that there is reduced privilege on custom tasks that the deployer of a language model chooses. This is important for selecting what privilege to grant from the perspective of the deployer.

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

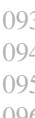

*Figure 11.* **MiniPile perplexity vs. MLP rank: SVD vs. NLPN** Qwen2.5-0.5B and Llama-3.2-1B under (1) SVD initialisation and (2) NLPN tuning on MiniPile. SVD shows catastrophic degradation; NLPN keeps stable perplexity across ranks via nested subspace training.

Figure 11 demonstrates that NLPN training enables stable rank reduction: perplexity remains close to full-rank performance across all tested ranks, with graceful degradation only at the lowest ranks. This property is essential for least-privilege deployment because it ensures that allocators can reliably predict utility at each privilege level, enabling principled policy optimization (Desideratum D3). In contrast, SVD truncation produces unpredictable behavior that would make privilege allocation unreliable in practice.