# OpenReview forum: "No More, No Less: Least-Privilege Language Models"
_ICML.cc/2026/Conference — ICML 2026 regular_

### Official Review · Reviewer_cczi · 2026-03-12

**Soundness:** 3
**Presentation:** 3
**Significance:** 3
**Originality:** 3
**Overall Recommendation:** 4
**Confidence:** 4

**Summary:**

This paper introduces the concept of least language models, drawing an analogy from the principle of least privilege in computer security to the development of language models. The core argument is that current LLM deployments expose the full internal capability of a model to every user and every request, whereas a principles deployment should restrict access to only the minimum internal computation needed to fulfill given request. The paper distinguishes their proposal from output-level controls (e.g., RLHF, constitutional AI, wrapper-based filtering) and activation steering by defining “privilege” as the reachable internal computation during the forward pass, a semantic definition that corresponds to restricting the model’s accessible function class rather than merely suppressing outputs.

**Compliance With Llm Reviewing Policy:**

Affirmed.

**Final Justification:**

This is a nice contribution and deserves consideration for acceptance.

**Key Questions For Authors:**

- The experiments are limited to ≤B parameters and synthetic MMLU tasks. Have the authors attempted any experiments on models at the 7B+ scale, or on naturalistic safety-relevant benchmarks (e.g., WMDP, WildTeaming, Do-Not-Answer)? Understanding how privilege-utility frontiers behave at scale is critical for the practical relevance of the paradigm. A positive answer, even preliminary results, would strengthen the paper’s significant claim.
- The probe experiment (Figure 15) shows that capacity is genuinely suppressed at low rank under the training distribution. However, has any evaluation been conducted under adversarial prompting (jailbreaks, prompt injection) or adaptive attacks that specifically target the allocator or attempt to recover suppressed capabilities at reduced rank? If an attacker can fool the allocator or find rank-specific exploits, the security narrative weakens considerably. Clarifying the threat model and the guarantees the framework cannot provide would help the contribution.
- The paper does not report GPU hours, memory overhead, or wall-clock time for the NLPN fine-tuning or the beam search optimization for selective suppression (Appendix E.2). Could the authors provide these numbers? Of the post-hoc training cost approaches, those of full fine-tuning run, the claimed advantages over retraining separate models are diminished.
- The selective suppression experiments (Figure 8, 14) show limited collateral degradation on 5 MMLU subjects. How does collateral damage
scale when attempting to suppress more fine-grained or entangled capabilities (e.g., suppressing knowledge of a specific chemical synthesis pathway without affecting general chemistry)? The granularity of suppression seems constrained by the MMLU taxonomy coarseness; results on more fine-grained benchmarks would be informative.
- The tested allocators are simple heuristics. Do authors have results or plans for training a learned allocator (e.g., a small classifier or reward
model) that maps request features to privilege levels? The paper’s framing naturally invites this, and understanding whether the allocator can learn effective policies over the privilege interface would clarify the practical viability of the full monitor-allocator-enforcer stack.

**Limitations:**

If the above questions and weaknesses do indeed indicate limitations of the approach, then the paper would do well to make this explicit with a dedicated discussion.

**Strengths And Weaknesses:**

Strengths:
- The paper makes genuinely novel contributions by incorporating the principle of least privilege from operating systems and access control
theory into the design space of LLM deployment. The insight of deployment control (wrappers, filters, activation steering) does not restrict the model’s function class but merely filters its outputs. It is well articulated and represents a meaningful shift in perspective relative to prior work. The taxonomy of deployment-time mechanisms (Figure 3) is particularly effective at clarifying this distinction from output wrappers, adaptive compute, and activation steering.
- The monitor allocator enforcer decomposition provides a principle and a modular abstraction for reasoning about deployment-time control. The four desiderata(D1-D4) are well motivated and provide the reader with a clear evaluation. The formalization of least-privilege inference as a constrained optimization problem that minimizes expected privilege, subject to a utility target for a good initiative.
- The paper is comprehensive for a paradigm-proposing paper. Five research questions are systematically addressed across multiple models,
tasks, allocation policies, and rank settings. The inclusion of the probing experiment (Appendix F, figure 15), demonstrating that probe recoverability collapses at low rank while prompt-based suppression merely masks output, is a strong piece of evidence distinguishing
NLPN-based privilege restriction from behavioral masking. The selective MMLU suppression results (Figures 7, 8, 14) with Benjamini-Hochberg FDR correction are rigorous and support the claim that block-level interventions can achieve targeted capability suppression with limited collateral damage.
- The extended related work (Appendix A) is thorough. The normative/epistemic/structural categorization (Appendix A.2) provides a
useful new lens for positioning deployment control mechanisms. The careful differentiation from slimmable networks and once-for-all models
is well tackled.
- The impact statement and Appendix B.4 acknowledges several important limitations: suppressed capabilities may be recoverable under adaptive prompting or fine-tuning; the framework could be misused for opaque censorship or discriminatory access; and no theoretical guarantees for suppression are provided. the honesty is commendable and helps ground the contribution.


Weaknesses:
- The three primary experimental tasks (Balanced Brackets, Length Comparison, and Contains Substring) are all algorithmically generated and controllable in difficulty. While this enables clean diagnosis, it leaves open the question of whether the privilege-utility frontiers are similarly well-behaved on naturalistic language tasks with less structured difficulty. The MMLU experiments are more naturalistic but limited to 5 STEM subjects, focus on small models, and test only selective suppressions rather than full privilege–utility frontiers. No experiments are conducted on open-ended generation, conversational settings, or safety-relevant benchmarks (e.g., WMDP, Wild Teaming) that would directly test the motivating use case of suppressing dangerous knowledge.
- All experiments use models in the 0.5-1B parameter range (Qwen2.5-0.5B, Pythia-1B, LLama3.22-1B). The paper’s motivating examples concern frontier-scale deployment (e.g., models providing bioweapons instructions to over a billion users), yet the gap between the experimental models and the deployment scenarios described in the introduction is large. Whether NLPNs scale to 7B+ models, where rank structures, knowledge distribution, and training dynamics differ qualitatively, is untested.
- The paper acknowledges the relationship between “rank” and “privilege” (i.e., semantic capability), but only empirically established and remains opaque. In Appendix B: “There is no reason to believe a single rank corresponds to a particular semantic capability.” This is a fundamental conceptual gap. The analogy to operating system privilege, where the privilege levels have precisely defined semantic (e.g., read/write/execute on specific resources), breaks down when the mapping between rank and capability is mediated by an opaque optimization procedure. In practice, reducing rank degrades “hard” tasks faster than “easy” ones, but the deployer cannot easily predict
which capabilities will be affected at a given rank without extensive empirical profiling.
- The paper describes fine-tuning on ~1B tokens of MiniPile to stabilize the rank interface (Figure 11). While this is lighter than full pretraining, it is not negligible and may limit adoption. The paper does not report training costs (GPU hours, memory overhead, wall-clock time) for the NLPN adaptation procedure. Furthermore, each new suppression target requires a beam search optimization over block-rank configurations, whose cost is also unreported.
- The five tested allocators (Full Privilege, Mini Rank, Static LP, progressive Incremental, Progressive Jump) are all simple heuristics based on calibrated uncertainty thresholds. The paper does not explore learned allocators, and the signal used is restricted to promptlevel information (s(x)=x). In motivating safety scenarios where a deployer wants to suppress dangerous chemical knowledge from unauthorized users, the allocator must distinguish query intent, which is itself a hard classification problem. The paper does not address how allocator errors (false positives/negatives) propagate through the system.
- The paper does not evaluate whether the privilege restrictions are robust to adversarial inputs. If an attacker can craft prompts that mislead the allocator into granting higher privilege, or if jailbreak style techniques can recover suppressed capabilities even at reduced rank, the security guarantees at the paradigm are weakened. The probing experiment (Figure 15) shows that probe accuracy collapses at low rank, but this is measured under the training distribution, and behavior under an adversarial distribution shift remains unexamined.

---

> ### Author Rebuttal · Authors · 2026-03-30
>
> Thank you for the careful and constructive review. We appreciate your very thoughtful review.
>
> **Major update summary**. In light of your review, we have: (a)added a NER task; (b) added larger models; \(c) ran **5 new experiments**, and (d) changed our manuscript to match your recommendations.
>
> # 1. Adding 5 new experiments, more models, more tasks
>
> (1) We agree that the original three main tasks are algorithmic and controllable in difficulty, and that more naturalistic evaluation is necessary. To address this directly, we added a new Named Entity Recognition verification task based on real English text with entity annotations. This task preserves our difficulty-binned evaluation setup while moving substantially closer to realistic language understanding. The results are consistent with the main paper: across Qwen2.5-0.5B, Pythia-1B, and Llama-3.2-1B, lowering rank still produces a smooth privilege–utility frontier, and harder examples degrade faster at lower privilege levels ([extension results](https://anonymous.4open.science/r/llc-AC18/EXTENSION_RESULTS.md) Sec. 1).
>
> We also extended the experiments to larger models. with Llama-3.2-3B and Qwen3-4B on the naturalistic NER task. The same qualitative pattern persists at larger scale. In fact, we think this strengthens the paper’s claim that the mechanism is not tied to one architecture or one model size.
>
> | Model | Params | Full Rank Acc. | Min Rank Acc. | Δ |
> |---|---|---|---|---|
> | Qwen2.5-0.5B | 0.5B | 93.9% | 80.0% | −13.9 pp |
> | Pythia-1B | 1.0B | 94.1% | 80.8% | −13.3 pp |
> | Llama-3.2-1B | 1.2B | 94.5% | 80.1% | −14.4 pp |
> | Llama-3.2-3B | 3.2B | 95.1% | 93.6% | −1.5 pp |
> | Qwen3-4B | 3.6B | 94.7% | 79.0% | −15.7 pp |
>
> To reiterate, we see the present paper as establishing the deployment object and showing that it has the right qualitative properties across multiple tasks, models, policies, and targeted suppression settings. The new naturalistic NER experiments and the additional 3B–4B results strengthen that case (even though  they do not exhaust the deployment questions you raise).
>
> # 2. We now highlight our main contribution better
>
> (2) We agree as well that rank is a coarse proxy for semantic capability restriction. **We view our primary contribution as the least-privilege deployment formulation** — defining privilege as reachable internal computation and decomposing control into monitoring, allocation, and enforcement. The rank subspace being one possible instantiation (but not the only one!). In fact, we hope that future work builds and finds better lesat-privilege architectures. We have made this more explicit in the main paper (Sec. 3-4).
>
> Also, **we conduct a probing study in Appendix F (Fig. 15) which shows that at low rank, probe recoverability collapses, whereas prompt-based suppression can hide outputs while task information remains linearly recoverable from activations**. We therefore agree that rank is a proxy, but the evidence suggests it is a proxy for internal capability reduction, not just for surface-level refusal behavior.
>
> # 3. We now provide practical deployment guidelines
> (3) Regarding practical deployment guidance for dynamic allocation and overhead, the paper already compares static and progressive policies in the privilege–utility–overhead space. The main pattern is that progressive incremental policies minimize average privilege but incur more inference passes, while jump-style policies trade slightly higher privilege for lower overhead. In the current results, progressive incremental policies cost roughly 2–3X overhead, whereas jump variants are closer to 1.3–1.5X, and the difference matters more when targeting very high utility levels. **We agree this should be stated more explicitly in the response and have updated our paper accordingly**.
>
> # 4. We now explain failure modes even better in the paper
>
> (4) On the concern about failure modes and uneven or unintended suppression, we agree this should be discussed more explicitly. **Interestingly, our empirical evidence so far suggests that suppression is not uniformly global: the block-level and targeted MMLU experiments show differential sensitivity across blocks and successful subject-level suppression** with limited off-target degradation (Fig. 12–14, Appendix E). At the same time, we do not claim a guarantee against unintended effects, and we believe the correct framing is that the current results show promising selectivity rather than a complete solution to deployment robustness.
>
> # Thank you
> Thank you for your feedback: you have helped us improve the paper. **In light of your review, we have: (a) added a NER task; (b) added larger models; \(c) ran **5 new experiments**, and (d) changed our manuscript to match your recommendations.**
>
> If you feel the paper, with the new experiments and framing, is in better shape, we would like to kindly encourage you to consider increasing you score.

---

> > ### Author Rebuttal · Reviewer_cczi · 2026-04-02
> >
> > Thank you for thoroughly addressing my comments, and the paper was already in nice shape.

---

> > > ### Author Response · Authors · 2026-04-03
> > >
> > > Dear reviewer cczi,
> > >
> > > Thanks for your comments:) As per before, you've helped us strenghten the paper. If you feel the paper, with the new experiments and framing, has addressed your concerns, we would like to kindly encourage you to consider increasing you score to an "Accept".

---

### Official Review · Reviewer_ztGY · 2026-03-13

**Soundness:** 3
**Presentation:** 4
**Significance:** 2
**Originality:** 3
**Overall Recommendation:** 4
**Confidence:** 3

**Summary:**

The authors argue for a new paradigm of safety control via providing least-privileged language models for users. The authors provide 3 layers: monitoring of inputs to determine privileges, a decision rule to map signals to a privilege setting, and an inference-time mechanism to implement the control. Concretely, the authors propose enforcing privilege by reducing appropriate rank of the parameters, thereby reducing utility.  The authors also note that rank reduction disproportionately hurts difficult instances versus easy ones.

**Compliance With Llm Reviewing Policy:**

Affirmed.

**Final Justification:**

I am maintaining my score and recommending weak acceptance. The paper has a unique angle and has a proof of concept implementation of limiting privilege by rank limitation. The authors have participated in the rebuttal period thoroughly and I believe this paper's merits outweigh its limitations.

**Key Questions For Authors:**

- Would tuning for rank reduction be an empirical question for each domain when there are fuzzy boundaries between dual-use and benign knowledge? How scalable would this be?

**Limitations:**

yes

**Strengths And Weaknesses:**

Strengths:
- Interesting framework of considering LLM access with a given base model
- Good presentation of idea, method, and experiments
- Good discussion around assumption of providing full capabilities to each user, and current concrete gaps in defenses

Weaknesses:
- The motivation for least-privilege language models is strongest for safety settings where certain capabilities is dual-use and contextually dangerous. The evaluations, though comprehensive in terms of models evaluated and block control, do not cover higher stake domains such as dual-use knowledge in biosecurity and/or cybersecurity.

---

> ### Author Rebuttal · Authors · 2026-03-30
>
> Dear Reviewer ztGY,
>
> Thank you for the thoughtful review and for highlighting an important scope question.
>
> **Major update summary**. In light of your review, we have: (a) added new tasks; (b) added larger models; (c) ran 5 new experiments, and (d) changed our manuscript to match your recommendations.
>
> # 1. We have improved our least-privilege explanation in the paper
>
> (1) First, we agree that the motivation for least-privilege language models is especially compelling in high-stakes dual-use settings, and that the original submission would be stronger with evidence beyond controlled tasks. **Our main idea is that the paper contributes both a general deployment-time framework and one concrete enforcement mechanism.** The framework itself is not specific to any one domain: it formalizes least-privilege control through the monitor–allocator–enforcer decomposition and defines privilege as reachable internal computation *(Sec. 3, Fig. 1)*. The role of the current experiments is to test whether this deployment interface has the right properties — precise reversible control, monotone privilege–utility trade-offs, and selective suppression — before specializing it to any one safety domain.
>
> # 2. We have added more tasks, experiments, more models, to validate our approach.
>
> (2) We also agree that broader empirical grounding matters. To address this concern, we extended the evaluation to a more naturalistic language setting by adding a Named Entity Recognition (NER) verification task built from real English sentences with entity annotations. This moves beyond fully synthetic algorithmic tasks while preserving the same rank-controlled evaluation protocol. Across Qwen2.5-0.5B, Pythia-1B, and Llama-3.2-1B, we observe the same pattern as in the main paper: **reducing privilege yields a smooth degradation frontier, and harder examples degrade earlier than easier ones** ([extension results](https://anonymous.4open.science/r/llc-AC18/EXTENSION_RESULTS.md) Sec. 1).
>
> We additionally extended the study to larger models in the 3B–4B range. On Llama-3.2-3B, accuracy remains high even under a 16× rank reduction (95.1% to 93.6%), while Qwen3-4B shows a steeper but still monotone decline (94.7% to 79.0%). We view these results as important evidence that the privilege–utility structure is not confined to sub-1B models, even though extending to 7B+ remains future work due to compute constraints.
>
> We cannot paste, unfortunately, results here, but we encourage you to look at our summarized extension results where we have visualized the model performance. Here is one summary table of different models on the aforementioned NER task:
>
> | Model | Params | Full Rank Acc. | Min Rank Acc. | Δ |
> |---|---|---|---|---|
> | Qwen2.5-0.5B | 0.5B | 93.9% | 80.0% | −13.9 pp |
> | Pythia-1B | 1.0B | 94.1% | 80.8% | −13.3 pp |
> | Llama-3.2-1B | 1.2B | 94.5% | 80.1% | −14.4 pp |
> | Llama-3.2-3B | 3.2B | 95.1% | 93.6% | −1.5 pp |
> | Qwen3-4B | 3.6B | 94.7% | 79.0% | −15.7 pp |
>
> We can see the performance is highly suppreseed via min rank, and we observe similar behavior in other experiments as well, in addition to those already in the paper.
>
> # 3. We have expanded our explanation of least-privilege architectures in other domains
>
> (3) On the question of whether tuning rank reduction would be an empirical question for each domain, especially when boundaries between benign and dual-use knowledge are fuzzy: yes, we agree that this is fundamentally domain-dependent. In our formulation, that is exactly why the paper separates the problem into monitoring, allocation, and enforcement. The enforcer provides a general internal control surface, while the allocator can be adapted to the deployment context, the user role, or the risk model (Sec. 3.2). In other words, we do not assume a single universal privilege policy across all domains; rather, we propose a deployment mechanism that makes such policies possible in a single model interface.
>
> # 4. We have clarified the scope of our claims
>
> (4) We also want to be careful about what we are and are not claiming. We are not claiming that the current paper already solves biosecurity or cybersecurity deployment. The claim is narrower: that least-privilege can be made into a meaningful deployment-time control variable over internal computation, and that NLPNs provide one concrete, reversible, shape-preserving way to realize that interface. The paper already states the least-privilege paradigm is broader than safety alone and that its current purpose is to establish the viability of this new deployment object (Sec. 1, Sec. 3).
>
> # Thank you
>
> Thank you for your feedback: you have helped us improve the paper. **In light of your review, we have: (a) added a NER task; (b) added larger models; (c) ran 5 new experiments, and (d) changed our manuscript to match your recommendations.**
>
> If you feel the paper, with the new experiments and framing, is in better shape, we would like to kindly encourage you to consider increasing you score.

---

> > ### Author Rebuttal · Reviewer_ztGY · 2026-04-03
> >
> > Thank you for the rebuttal and the clarification around the paper's claims, as well as additional experiments. The paper has a unique angle considering least-privilege deployment time control. I am less sure about the practical application, and I understand that high stake tasks are beyond the scope of this paper. I think this paper proposes an interesting research direction that is meaningful for the community. Because my reservations about practicality still holds, I'm maintaining my weak acceptance score.

---

> > > ### Author Response · Authors · 2026-04-03
> > >
> > > Dear Reviewer ztGY,
> > >
> > > Thank you for the kind response and reviews.
> > >
> > > Given that our work (a) advances and explains not just the **practical** (how can we reduce privilege), but also **foundational** machine learning work (how can we in principle obtain least-privilege language models theoretically), **and** that (b) we have significantly expanded the experimental setup to include new tasks, larger models, 5 new experiments, and incorporated your suggestions to the manuscript, could we kindly ask you to consider increasing your score in light of this information?
> > >
> > > In fact, your comment has prompted us to include one more paragraph in the discussion section on how an agenda that can enable the practical applications of least-privilege networks. We'd be delighted to consider other changes if you think they might merit the paper and benefit the community.

---

### Official Review · Reviewer_GAai · 2026-03-15

**Soundness:** 3
**Presentation:** 3
**Significance:** 2
**Originality:** 3
**Overall Recommendation:** 4
**Confidence:** 3

**Summary:**

This paper proposes least-privilege language models as a deployment-time control framework for LLMs. Instead of exposing the same full-capability policy to all users and relying mainly on alignment or output filtering, the paper argues that systems should expose only the minimum internal capability needed for each request. To formalize this, the paper introduces a monitor–allocator–enforcer decomposition and defines privilege in terms of reachable internal computation during inference.

As a concrete instantiation, the paper proposes Nested Least-Privilege Networks (NLPNs), which reparameterize selected transformer linear layers into rank-indexed low-rank factors, enabling inference-time control through rank truncation. Experiments suggest that reducing privilege leads to smooth utility degradation, with harder tasks degrading earlier than easier ones; dynamic allocation policies can achieve better utility–privilege trade-offs than simple static policies; and some degree of selective task/subject suppression is possible with limited collateral damage.

Overall, the paper contributes both a new conceptual framing for deployment-time model control and a concrete mechanism with initial empirical evidence of feasibility.

**Compliance With Llm Reviewing Policy:**

Affirmed.

**Ethical Review Flag:**

Flag this paper for an ethics review.

**Final Justification:**

I keep my score.

**Key Questions For Authors:**

1. The current framework instantiates privilege mainly through rank reduction. How specific are the results to this particular enforcement surface, and how much do the authors expect the least-privilege framework to generalize beyond rank-based control?
Effect on evaluation: A clearer answer would strengthen the paper’s conceptual significance.

2. The selective suppression results are interesting. Could the authors clarify how robust these effects are across seeds, model sizes, and broader or more adversarial evaluations?
Effect on evaluation: This would increase my confidence in the generality of the empirical findings.

3. Dynamic allocation improves average privilege, but can add inference overhead. Could the authors provide more guidance on when progressive policies are preferable to static ones in realistic deployment settings?
Effect on evaluation: This would strengthen the paper’s practical impact.

**Limitations:**

The paper presents a promising new deployment perspective, but the current evidence still seems somewhat preliminary. Much of the evaluation is on controlled or benchmark-style tasks, so it remains unclear how well the framework transfers to realistic open-ended deployment settings. In addition, the current mechanism defines privilege through rank-based restriction, which is technically coherent but may still be a relatively coarse proxy for semantic capability restriction. Finally, the paper could discuss failure modes more explicitly, especially the risk of uneven or unintended capability suppression under deployment.

**Strengths And Weaknesses:**

Strengths

1. The paper introduces a clear and novel deployment-time perspective: controlling internal capability exposure rather than only post-hoc output behavior. This is a meaningful conceptual contribution.

2. The monitor–allocator–enforcer decomposition is clean and useful. It provides a principled way to think about request-conditioned deployment-time control.

3. The paper includes a concrete mechanism, NLPNs, rather than remaining purely conceptual. The rank-indexed construction makes the least-privilege idea operational and testable.

4. The empirical results support several important properties needed for the framework to be useful, including monotone degradation with reduced privilege and differential sensitivity of easy vs. hard tasks.

5. The policy comparison is valuable: dynamic allocation appears able to meet target utility with lower average privilege than simple static alternatives.

6. The block-level and subject-level suppression results are particularly interesting, since they suggest that privilege control may be made more selective than a single global capacity knob.

Weaknesses
1. The empirical evidence is promising but still somewhat preliminary. Much of the evaluation appears to rely on controlled or benchmark-style tasks, so the extent to which the framework transfers to realistic open-ended deployment settings remains unclear.

2. The choice of rank reduction as the operationalization of privilege is technically coherent, but it may still be a relatively coarse proxy for semantic capability restriction. It is not yet clear how general this privilege axis is across broader model behaviors.

3. The selective suppression results are intriguing, but they would be stronger with broader validation across more models, seeds, and more natural or adversarial evaluation settings.

4. The policy comparisons are useful, though the baseline policy space is still limited. Comparisons to a wider set of adaptive inference/control baselines would strengthen the empirical case.

5. The paper is conceptually dense. While generally well organized, some of the formal framing around privilege, reachability, and function class restriction may be difficult for readers to connect to the practical mechanism on first pass

---

> ### Author Rebuttal · Authors · 2026-03-30
>
> Dear Reviewer GAai,
>
> Thank you for the constructive questions.
>
> **Major update summary**. In light of your review, we have: (a) added a new task; (b) added larger models; (c) ran 5 new experiments, and (d) changed our manuscript to match your recommendations.
>
> # 1. We have improved our discussion on enforcement surfaces.
>
> (1) **Other enforcement surfaces.** The current empirical results are specific to the rank-based NLPN enforcement surface. We chose rank reduction because it gives a first mechanism with the properties we needed: it is post-hoc, shape-preserving, reversible, and exposes a smooth privilege–utility frontier inside the forward pass. The latter quality also makes targeted-suppression optimisation more tractable. That said, **the least-privilege framework itself is not tied to rank. In the paper, least privilege is defined at the level of the monitor–allocator–enforcer decomposition and restriction of reachable internal computation** (which is our prinary contribution); NLPNs are presented as one concrete instantiation. We will make this more explicit.
>
> # 2. We have clarified selective suppression.
>
> (2) **Selective suppression.** In the current paper, the central empirical claim is that selective suppression is feasible: sparse block/projection interventions can reduce performance on targeted subjects while preserving utility on others, as illustrated in Figs. 7–8 and related appendix results. **Across multiple random seeds, we observe that the effect is qualitatively stable—targeted suppression with limited collateral degradation recurs, and for a given suppression task and fine-tuned model, different seeds yield similar, but not identical, block–projection–suppression patterns**. This is consistent with our observation that multiple sparse configurations can realise similar selective-suppression outcomes, rather than there being a single canonical block pattern. We will make this more expliciti n the paper.
>
> Simultaneously, we do not want to overclaim generality beyond what is currently shown. The present selective-suppression evidence is specific to the models and setups evaluated in the paper, and we have not yet established that the same block-level configurations transfer across model families or model sizes. Likewise, we have not yet performed a dedicated adversarial recoverability evaluation for these selective interventions. This is also consistent with the paper’s stated limitations, which note that suppressed capabilities may be recoverable under adaptation and stronger guarantees remain open.
>
> # 3. We have improved our explanation on dynamic allocation policies
>
> (3) **Progressive vs Static policies.** We use the policies as examples to showcase that there are *in principle* different policy mechanisms that can achieve different outcomes. In our paper, progressive policies are preferable precisely when privilege minimisation is itself an important deployment objective, and when the request distribution is heterogeneous enough that a single static setting would over-allocate privilege on many easy cases. In that setting, progressive policies achieve a better privilege–utility trade-off than static LP, at the cost of extra passes on hard cases. On the other hand, static policies are preferable when latency or throughput budgets dominate, or when the workload is sufficiently homogeneous that one calibrated rank already meets the target utility across most requests. This is also visible in our results: **at low utility targets, many policies converge to similarly low privilege with minimal overhead; and on some model–task pairs, the same low rank already suffices, so progressive escalation offers little benefit**.
>
> **Most importantly, this showcase that we can design policies that support different privilege levels**. We have now made this more explicit in the writeup.
>
> We would therefore summarise the deployment guidance as follows. Use Static LP when one additional calibration step is acceptable but repeated inference is not, or when the workload is stable and near-homogeneous. Use Progressive (Incremental) when minimizing average privilege is the primary objective and modest additional latency is acceptable. Use Progressive (Jump) when some adaptivity is desirable but there is a tighter cap on inference passes, since it trades some privilege efficiency for lower overhead. We will make this decision rule explicit in the revision, since we agree that practical guidance is important for interpreting the policy comparisons.
>
> # Thank you
>
> Thank you for your feedback: you have helped us improve the paper.
>
> If you feel the paper, with the new experiments and framing, is in better shape, we would like to kindly encourage you to consider increasing you score.

---

> > ### Author Rebuttal · Reviewer_GAai · 2026-04-03
> >
> > My concerns have been addressed. I keep my score.

---

> > > ### Author Response · Authors · 2026-04-03
> > >
> > > Dear Reviewer GAai,
> > >
> > > Thanks for your response. Given that (a) your reviews have been addressed in full; and (b) we have significantly expanded the experimental setup to include new tasks, larger models, 5 new experiments, and incorporated your suggestions to the manuscript, could we kindly ask you to consider increasing your score accordingly?

---

### Official Review · Reviewer_Gtxe · 2026-03-24

**Soundness:** 3
**Presentation:** 3
**Significance:** 3
**Originality:** 3
**Overall Recommendation:** 4
**Confidence:** 3

**Summary:**

The paper proposes a new level of security in deployed LLMs called least privilege where privilege is defined as the reachable internal computation. This least privilege enforces and limits the accessible function class. A nested least privilege network is also provided which provides a smooth utility and privilege tradeoff through rank indexed intervention.

**Compliance With Llm Reviewing Policy:**

Affirmed.

**Key Questions For Authors:**

Merged with weakness

**Limitations:**

yes

**Strengths And Weaknesses:**

Strengths:
1) Novel framing: capability control instead of output control
2) The NLPN provides smooth controllable knob


Weaknesses:
1) From a security perspective I am unable to understand the intuition as to why necessarily low rank --> less dangerous knowledge and high rank--> more dangerous knowledge. It is a mathematical proxy but no theoretical justification exists.

---

> ### Author Rebuttal · Authors · 2026-03-30
>
> Dear Reviewer Gtxe ,
>
> Thank you for your feedback.
>
> We agree that low rank does not, by itself, imply “less dangerous knowledge,” nor do we assume a one-to-one correspondence between rank and any semantic capability. The paper’s claim is narrower: rank serves as an **operational proxy for privilege**, where privilege is defined as the set of **reachable forward-pass computations**. Under Definition 3.1, lowering privilege shrinks that reachable computation set; in our NLPN instantiation, Section 4.1 formalises this through the nested-subspace property $\(\mathrm{Im}(W(g)) \subseteq \mathrm{Im}(W(g+1))$, so decreasing $g$ restricts the accessible function class.
>
> **Because low rank implies lower reachable internal computation, we can guarantee that there are some computations that are never reached, by definition**. We show empirically that this is the case in a number of our studies. Specifically, our experiments (e.g. **Fig 4, Fig 5, Fig 6, Fig 7**) directly show that it is possible to reduce rank and obtain worse performance (lower privilege) on desired levels, and **we even show that we can suppress capabilities empirically without removing other harmful capabilities** (Fig 8.). We think this is clear justification for this claim, empirically. In fact, we have **now added more experiments to highlight this claim**, with a named entity recognition task; (b) added larger models; and (c) ran 5 new experiments. All of these further confirm our insights. In fact, the security motivation is precisely why we distinguish privilege allocation from wrapper-level controls or compute-only optimisation: those may alter outputs or cost, but they do not necessarily change which internal computations remain reachable, whereas our mechanism is designed to do so.
>
> We will revise the text to make this distinction more explicit and to avoid any implication that rank is a semantic danger measure in itself: Section 3.1 and the discussion will state explicitly that rank is not a semantic danger score, only a monotone enforcement variable over reachable computation.
>
> # Thank you
>
> If you feel we have addressed your concerns and that there are no outstanding issues, we would like to kindly encourage you to consider increasing you score.

---

> > ### Author Rebuttal · Reviewer_Gtxe · 2026-04-04
> >
> > Thanks for the resolution

---

> > > ### Author Response · Authors · 2026-04-04
> > >
> > > Dear reviewer Gtxe,
> > >
> > > Thanks for your acknowledgement. If your concerns have been fully addressed, please consider adjusting your score accordingly. If not, we'd be happy to see what other remaining clarifications we can make on our end to improve the quality of the paper.
> > >
> > > Thanks for your input in the review process.

---

### Decision · Program_Chairs · 2026-04-30

**Decision:**

Accept (regular)

**Comment:**

The authors unanimously concluded that the paper provides a unique and interesting angle for deployment-time control in LLMs. None of the reviewers strongly advocated for the paper because the evaluation was more "proof of concept" on toy settings. Despite that limitation, I believe this paper provides new ideas that the community should consider and debate/discuss/extend.